# Learning Asymmetric Phase Dynamics via Clinically-Guided Spatiotemporal Fusion

## Abstract

Hepatocellular carcinoma is a leading cause of cancer-related mortality, where accurate tumor characterization is crucial for guiding treatment. In clinical consensus, multiphase contrast-enhanced computed tomography (CECT) is indispensable: arterial (ART), portal venous (PV), and delayed (DL) phases jointly depict tumor vascularization, perfusion heterogeneity, and fibrotic evolution, forming the very basis of radiologists' diagnostic reasoning. *Surprisingly, despite this well-established clinical value*, most AI models still rely on ① single-phase inputs or naive stacking of multiphase scans, ② ignoring temporal hemodynamics and lacking interpretability. To bridge this gap, we present **Clinically-Guided Spatiotemporal Deep Fusion Network** (`CSF-Net`), the first framework that explicitly embeds radiological knowledge into multiphase modeling. `CSF-Net` incorporates three synergistic components: the multi-phase clinical-quantitative synergy branch (**MCQS**) for phase-specific encoding, the temporal-aware local feature refinement module (**TLFR**) for perfusion dynamics, and the query-interaction enhancement fusion module (**QIEF**) for cross-phase alignment. By aligning AI modeling design with radiologists' logic, `CSF-Net` establishes a clinically grounded interpretability paradigm, which inevitably yields superior performance gains. Extensive experiments on two CECT benchmarks, PLC-CECT and MPLL, demonstrate that `CSF-Net` achieves state-of-the-art performance. Our codes are available at https://anonymous.4open.science/r/ICLR26_CSF-Net-63E3/.

## 1 Introduction

Liver cancer, particularly hepatocellular carcinoma (HCC), remains one of the leading causes of cancer-related mortality worldwide (Zhao & Li, 2025; Hu et al., 2024; Lyu et al., 2021). Accurate assessment of tumor burden and biological behavior is crucial for guiding treatment strategies and improving patient outcomes (Shaker et al., 2024; Zheng et al., 2022; Valanarasu et al., 2021). In clinical practice, contrast-enhanced computed tomography (CECT) is the primary imaging modality for liver tumor diagnosis (Huang et al., 2024), staging, treatment response evaluation, and longitudinal follow-up (Song et al., 2025; Wang et al., 2025). Compared with single-phase static imaging, multiphase CECT captures the dynamic evolution of tumor perfusion, providing a richer source of biological information for radiomics and deep learning models (Poetter-Lang et al., 2023; Zhou et al., 2021; Stocker et al., 2020).

CECT typically comprises **four phases**: non-contrast (NC), arterial (ART), portal venous (PV), and delayed (DL) (Liu et al., 2024; Poetter-Lang et al., 2023; Elsayes et al., 2017). Each phase reflects distinct perfusion states and offers complementary clinical and biological cues (Poetter-Lang et al., 2023; Yang et al., 2019). ♣ **NC** depicts baseline attenuation (HU values), enabling detection of fat infiltration, fibrosis, and tissue density variations (Xu et al., 2022a). ♥ **ART** reveals early arterial enhancement driven by neovascularization and correlates strongly with microvascular density (MVD) (Haller et al., 2025), making it sensitive to hypervascular tumors such as HCC (Albano et al., 2021; Yang et al., 2019). ♦ **PV** captures the peak of hepatic parenchymal perfusion (Menezes et al., 2025), yielding the best contrast between tumors and surrounding tissue for global heterogeneity analysis (Albano et al., 2021; Yang et al., 2019). ♠ **DL** characterizes contrast clearance and retention, reflecting extracellular extravascular space (EES) properties associated with fibrotic reactions, capsule formation, and necrosis (Fujita et al., 2017).

Together, these four phases form a temporally continuous hemodynamic chain, from angiogenesis to contrast distribution and washout (Liu et al., 2024; Al-Battal et al., 2024; Layton & Lapsia, 2023; Zhou et al., 2021). Despite the abundant clinical and biological information embedded in this sequence, existing deep learning approaches remain limited in clinical interpretability (Zhang et al., 2023; Xu et al., 2021; Zhang et al., 2021b). They often lack clinical priors and rely solely on statistical correlations, failing to capture temporal dependencies of vascular perfusion dynamics (Qiao et al., 2024; Xu et al., 2022b; Huang et al., 2024). Most studies further focus exclusively on the PV phase, neglecting the complementary roles of ART and DL, and few methods explicitly model global temporal dynamics across phases (Qiao et al., 2024; Xu et al., 2022b; 2021). Consequently, current deep learning-based methods optimize for algorithmic accuracy *rather than producing clinically meaningful and biologically aligned insights*.

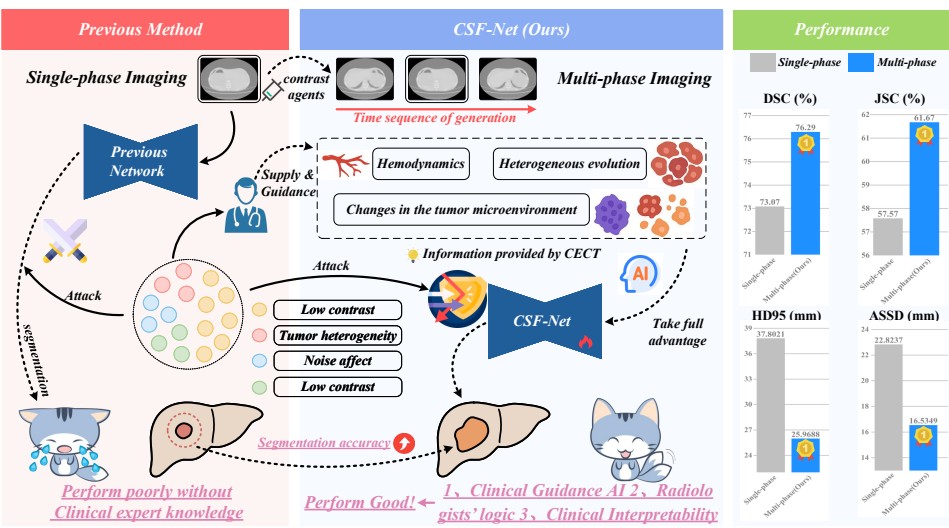

Figure 1: `CSF-Net` bridges clinical knowledge and AI design by modeling multiphase hemodynamics (ART, PV, DL). Unlike single-phase methods, this clinically grounded paradigm aligns AI modeling with radiologists' logic, improves interpretability, and naturally delivers state-of-the-art liver tumor segmentation performance.

To address these limitations, we propose **Clinically-Guided Spatiotemporal Deep Fusion Network (CSF-Net)**, the first framework that explicitly integrates radiological priors into spatiotemporal modeling of multiphase CECT. `CSF-Net` incorporates domain knowledge of tumor hemodynamics into network design: a spatial branch encodes phase-specific textural and anatomical cues while maintaining liver structural consistency, and a temporal branch explicitly models the dynamic perfusion progression from ART → PV → DL to capture inter-phase dependencies. As shown in Figure 1, by aligning deep modeling with clinical knowledge and expert reasoning logic, this design ensures interpretability and inevitably yields superior performance.

The main contributions of our work are as follows:

❶ We introduce `CSF-Net`, the first clinically guided spatiotemporal fusion framework for multi-phase CECT, bridging radiological expertise with deep feature learning.

❷ We design three complementary modules for spatiotemporal fusion of ART, PV, and DL: multi-phase clinical-quantitative synergy branch (**MCQS**) for phase-specific encoding, temporal-aware local feature refinement module (**TLFR**) for modeling perfusion dynamics, and query-interaction enhancement fusion module (**QIEF**) for multi-feature alignment and integration.

❸ We conduct comprehensive evaluations on two representative multiphase CT benchmarks (PLC-CECT and MPLL), where `CSF-Net` achieves **state-of-the-art** performance (+**3.22**% & +**1.09**% higher than the best single-phase & multi-phase baseline) and establishes a clinically grounded paradigm for multiphase tumor analysis.

## 2 RELATED WORKS

**Why Multiphase Matters: Clinical Perspectives on CECT.** Conventional single-phase CT offers only static anatomical snapshots, often insufficient for characterizing vascular dynamics or sub-

tle lesions. By contrast, multiphase CECT captures arterial (ART), portal venous (PV), and delayed (DL) hemodynamics (Kharga et al., 2025; Åhlström et al., 2025; Gyselaers, 2023; Nguyen et al., 2022; Uhm et al., 2022; Xu et al., 2022b), enabling radiologists to track tumor perfusion, delineate boundaries, and detect atypical enhancement patterns. Clinical studies confirm that ART and PV fusion improves tumor boundary delineation and mimics the radiologist's cross-validation process (Guo et al., 2022; Xu et al., 2021; Zhang et al., 2021b; Zaky et al., 2017), while ART, PV, and DL combinations provide complementary cues for vascularization and proliferative activity (Song et al., 2025; Zhu et al., 2022). Beyond pairwise fusion, studies progressively incorporated four-phase frameworks aligned with physiological hemodynamics to simulate clinical interpretation logic (Qiao et al., 2024; Huang et al., 2024; Stankovic et al., 2012). Collectively, these studies confirm that multi-phase fusion not only improves lesion conspicuity and diagnostic confidence but also aligns closely with radiologists' reasoning strategies, thereby reducing dependence on invasive procedures.

**AI at the Crossroads: Unlocking the Power of Multiphase Fusion.** Early AI models treated multiphase (ART, PV, and DL) scans as stacked channels, neglecting the intrinsic heterogeneity across phases (Zhu et al., 2022; Xu et al., 2021; Zhang et al., 2021b). Such approaches captured gross anatomical cues but often failed to leverage phase-specific vascular signatures or temporal progression patterns. Recent advances introduced phase-aware fusion strategies (Liu et al., 2024; Huang et al., 2024), where voxel- and pixel-level attention emphasized informative regions and aggregated complementary enhancement cues, while inter-phase attention aligned subtle temporal variations to capture dynamic enhancement trends (Liu et al., 2024; Qiao et al., 2024). In parallel, efforts addressing *phase misregistration and distributional shifts* introduced stage-wise alignment to reduce motion-induced mismatches (Qiao et al., 2024; Zhang et al., 2023; Al-Battal et al., 2024) and domain-adaptive fusion to mitigate intensity inconsistencies across phases (Ni et al., 2024).

In summary, existing studies underscore the indispensability of multiphase CECT and advances in AI-based fusion, yet no unified framework fully exploits phase-specific characteristics with explicit clinical interpretability. Unlike prior works, we first validate the clinical-to-AI bridge in single-phase experiments, and then introduce the Clinically-Guided Spatiotemporal Deep Fusion Network, where aligning AI reasoning with radiologists' logic makes performance gains inevitable.

## 3 PRELIMINARY

In this section we introduce the notations and formal foundations used throughout the paper, and provide mathematical definitions that underpin `CSF-Net`'s multi-phase encoding, directed feature propagation, local enhancement, spatio-sequential decoding, and final optimization objective. A detailed pseudocode description of the overall procedure is provided in Appendix A.

**Notations.** Let the set of contrast-enhanced CT images for a patient be $\mathcal{X} = \{\boldsymbol{X}^{(ART)}, \boldsymbol{X}^{(PV)}, \boldsymbol{X}^{(DL)}\}$, where each $\boldsymbol{X}^{(i)} \in \mathbb{R}^{H \times W \times c_{in}}$. We denote by $\boldsymbol{X}^{(\Sigma)} = \text{Concat}_c \, \boldsymbol{X}^{(ART)}, \boldsymbol{X}^{(PV)}, \boldsymbol{X}^{(DL)}$ the channel-wise reference fusion. Feature encoders map images to latent maps $\boldsymbol{F}^{(i)} \in \mathbb{R}^{H' \times W' \times C'}$. The directed clinical graph is $\mathcal{G} = (\mathcal{V}, \mathcal{E})$ with nodes $\mathcal{V} = \{\text{ART}, \text{PV}, \text{DL}, \Sigma\}$. Finally, let $Y \in \{0, 1\}^{H \times W}$ be the ground-truth segmentation mask and $\hat{Y}$ the network output.

**Multi-phase shared encoding & Directed clinical-prior message passing.** Each phase is encoded by a local-to-global encoder (shared arch., phase-specific params). A single-line encoding that combines depthwise, pointwise spatial processing, multi-head attention and a residual projection. We formalize clinically-informed propagation as attentioned graph message passing with priority bias and learnable gating, the $r_{j \rightarrow i}$ encodes clinical-priority bias, $\phi_{j \rightarrow i}$ is a learnable gate and $\sigma(\cdot)$ is sigmoid, the updated node feature is $\widetilde{\boldsymbol{F}}^{(i)} = \phi_i(\boldsymbol{F}^{(i)} W_i + \boldsymbol{M}^{(i)})$:

$$\boldsymbol{M}^{(i)} = \sum_{j \in \mathcal{P}(i)} \alpha_{j \rightarrow i} \big( \boldsymbol{F}^{(j)} W_{j \rightarrow i} + \boldsymbol{b}_{j \rightarrow i} \big) \odot \sigma(\phi_{j \rightarrow i}). \tag{1}$$

**Local separable enhancement (LSE) for boundary preservation.** To keep fine-grained boundary and texture cues we apply a depthwise, pointwise residual block with group normalization, the $\alpha, \beta$ are learnable scales, $\boldsymbol{\kappa}$ a small spatial kernel and $*$ denotes convolution, residual scaling and a lightweight squeeze-excitation term:

$$\boldsymbol{Y}^{(p)} = \text{GN}_g \Big( \boldsymbol{G}^{(p)} + \alpha \cdot \text{PW}(\text{DW}(\boldsymbol{G}^{(p)})) + \beta \cdot \text{SE}(\text{GAP}(\boldsymbol{G}^{(p)})) \Big). \tag{2}$$

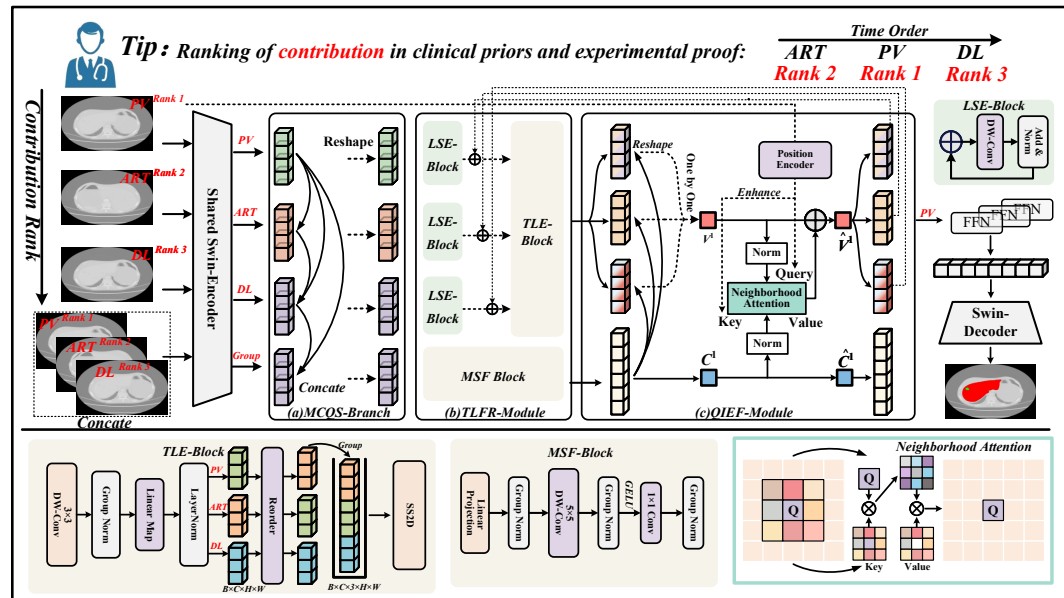

Figure 2: Overview of `CSF-Net`. (a) MCQS-Branch propagates multi-phase features with clinical priors; (b) TLFR-Module refines boundaries and aligns semantics for temporal modeling; (c) QIEF-Module fuses cross-phase features via positional attention and channel-wise enhancement.

We stack phase-specific low-level features in chronological order to form $\boldsymbol{S} \in \mathbb{R}^{T \times H' \times W' \times C'}$ and apply per-pixel multi-head temporal attention with learned biases and spatial-distance regularization:

$$\boldsymbol{A}_p^h = \text{Softmax}\left(\frac{\boldsymbol{Q}_p^h(\boldsymbol{K}_p^h)^\top + \boldsymbol{B}_p^h}{\sqrt{d_k}} + \lambda_{\text{tem}}\tanh\left(\boldsymbol{Q}_p^h\boldsymbol{R}(\boldsymbol{K}_p^h)^\top\right) + \rho\frac{\sum_{q \in \mathcal{N}(p)}\|\boldsymbol{r}_p - \boldsymbol{r}_q\|_2^2}{Z}\right), \quad (3)$$

where $\boldsymbol{r}_p$ is the spatial coordinate of pixel $p$, $\mathcal{N}(p)$ its neighborhood, and $\boldsymbol{B}_p^h$ a learnable bias table (RoPE/relative bias may be injected into $\boldsymbol{B}$).

## 4 METHODOLOGY

In order to fully utilize the complementary anatomical and pathological information provided by each phase of CECT, we propose `CSF-Net`, which consists of: ❶ Multi-Phase Clinical-Quantitative Synergy Branch (MCQS-Branch). ❷ Temporal-Aware Local Feature Refinement Module (TLFR-Module). ❸ Query-Interaction Enhancement Fusion Module (QIEF-Module). This section will introduce these components in turn.

### 4.1 MCQS-BRANCH

**Phase Ranking and Multi-Branch Input.** As shown in Figure 2 (a), we construct four input branches corresponding to different clinical phases. Given contrast-enhanced CT images from portal venous (PV), arterial (ART), and delayed (DL) phases, we define: $\boldsymbol{\mathcal{X}} = \left\{\boldsymbol{X}^{(PV)}, \boldsymbol{X}^{(ART)}, \boldsymbol{X}^{(DL)}\right\}, \boldsymbol{X}^{(i)} \in \mathbb{R}^{H \times W \times c_{in}}, i \in \{\text{PV}, \text{ART}, \text{DL}\}$. Furthermore each input image is pre-processed / projected with a light conv + normalization branch. To provide a reference fused view, we additionally construct:

$$\boldsymbol{X}^{(\Sigma)} = \Psi\left(\oplus_i\left(\boldsymbol{W}_i * \boldsymbol{X}^{(i)} + \boldsymbol{b}_i\right)\right) \in \mathbb{R}^{H \times W \times 3c_{in}}, \quad i \in \{\text{PV}, \text{ART}, \text{DL}\}, \quad (4)$$

where $\Psi(\cdot)$ denotes a channel-mixing projection (e.g. 1×1 conv + BN + activation).

**Shared Swin Transformer Feature Encoding.** The four input branches are fed into Swin Transformer encoders $\boldsymbol{\varepsilon}_{\boldsymbol{swin}}$ with a shared architecture but different weights for local-to-global feature extraction:

$$\boldsymbol{F}^{(i)} = \varepsilon_{\text{swin}}\left(\zeta(\boldsymbol{X}^{(i)}) + \boldsymbol{W}_q\boldsymbol{X}^{(i)}(\boldsymbol{W}_k\boldsymbol{X}^{(i)})^\top\boldsymbol{W}_v\boldsymbol{X}^{(i)}\right), \quad i \in \{\text{ART}, \text{PV}, \text{DL}, \Sigma\}, \quad (5)$$

where $\zeta(\cdot)$ is a local-window normalization / residual operator and $\boldsymbol{W}_q, \boldsymbol{W}_k, \boldsymbol{W}_v$ are linear projections used to emphasize local-to-global interactions within the Swin blocks. This mechanism is particularly important for spatial structure alignment across images from different contrast phases.

**Feature Flow with Directed Propagation.** To explicitly encode clinical priors, we formalize the multi-phase interactions as a directed acyclic graph (DAG): $\mathcal{G} = (\mathcal{V}, \mathcal{E})$, $\mathcal{V} = \{\mathrm{PV}, \mathrm{ART}, \mathrm{DL}, \Sigma\}$, where the edge set $\mathcal{E}$ captures clinically valid propagation paths (e.g., $\mathrm{PV} \to \mathrm{ART} \to \mathrm{DL}$). Given encoded features $\{\boldsymbol{F}^{(i)}\}$, the propagation into node $i$ is defined as a regularized graph-attention message passing:

$$\boldsymbol{M}^{(i)} = \sum_{j \in \mathcal{P}(i)} \frac{\exp\left((\boldsymbol{q}_i \boldsymbol{U})(\boldsymbol{k}_j \boldsymbol{V})^\top / \sqrt{d} + \lambda\, r_{ij}\right)}{\sum_{j' \in \mathcal{P}(i)} \exp\left((\boldsymbol{q}_i \boldsymbol{U})(\boldsymbol{k}_{j'} \boldsymbol{V})^\top / \sqrt{d} + \lambda\, r_{ij'}\right)} \left(\boldsymbol{F}^{(j)} \boldsymbol{W}_{j \to i} + \boldsymbol{b}_{j \to i}\right), \quad (6)$$

where $\boldsymbol{q}_i = \mathrm{GAP}(\boldsymbol{F}^{(i)}) U_i$, $\boldsymbol{k}_j = \mathrm{GAP}(\boldsymbol{F}^{(j)}) V_j$ are projected queries/keys, $\boldsymbol{U}, \boldsymbol{V}$ are projection matrices, $r_{ij}$ is a clinical-priority bias (scalar) and $\lambda$ a scalar gating coefficient; $\mathcal{P}(i)$ denotes the parent set of $i$ in $\mathcal{G}$. The updated feature at node $i$ is obtained via residual message integration with learned mapping: $\widetilde{\boldsymbol{F}}^{(i)} = \phi_i\left(\boldsymbol{F}^{(i)} \boldsymbol{W}_i + \boldsymbol{M}^{(i)}\right)$, where $\phi_i(\cdot)$ denotes a 1×1 conv + BN + GELU non-linear mapping.

In practice, the graph is constructed such that PV acts as the global broadcaster with out-degree 3, ART propagates to DL and $\Sigma$, and DL only transmits to $\Sigma$. This ensures a hierarchically weighted feature flow, where strong phases (PV, ART) enrich weaker phases (DL), while avoiding uniform averaging. The final refined representations $\{\widetilde{\boldsymbol{F}}^{PV}, \widetilde{\boldsymbol{F}}^{ART}, \widetilde{\boldsymbol{F}}^{DL}\}$ are forwarded into the subsequent temporal modeling branch.

## 4.2 TLFR-MODULE

As shown in Figure 2 (b), after the directed information injection in the CQS-Branch, the phase-specific features are refined through local spatial, channel enhancement and per-channel linear mapping. This process ensures that, before being fed into SS2D, each phase feature preserves detailed boundary information while achieving a unified semantic representation.

**Local Separable Enhancement (LSE-Block).** Let $\boldsymbol{G}^{(p)} \in \mathbb{R}^{H \times W \times C}$ be the input for phase $p$. $DW : \mathbb{R}^{H \times W \times C} \to \mathbb{R}^{H \times W \times C_\ell}, PW : \mathbb{R}^{H \times W \times C} \to \mathbb{R}^{H \times W \times C_\ell}$ Define the depthwise separable block as composition of a depthwise spatial operator and a pointwise channel mixing: $\mathcal{S}(\boldsymbol{G}^{(p)}) \triangleq PW\left(DW(\boldsymbol{G}^{(p)})\right) \in \mathbb{R}^{H \times W \times C_\ell}$. To make the representation stable and explicitly control residual contribution, we introduce a learnable residual scaling $\alpha \in \mathbb{R}$ and group normalization with $g$ groups. Let $\boldsymbol{W}_{dw}$ and $\boldsymbol{W}_{pw}$ denote the depthwise and pointwise kernels, and $\boldsymbol{b}_{dw}, \boldsymbol{b}_{pw}$ their biases, together with explicit conv-bias terms:

$$\boldsymbol{Y}^{(p)} = GN_g\left(\boldsymbol{G}^{(p)} + \alpha \cdot \left(\boldsymbol{W}_{pw}(\boldsymbol{W}_{dw} * \boldsymbol{G}^{(p)} + \boldsymbol{b}_{dw}) + \boldsymbol{b}_{pw}\right)\right). \quad (7)$$

**Temporal-aware Low-level Enhancement Block (TLE-Block).** The LLFE-Block further focuses on supplementing low-level features and performing linear mapping. Specifically, depthwise separable convolution combined with Group Normalization is first employed to refine local boundary and texture information. Subsequently, a 1×1 channel-wise linear projection is applied to reorganize the features, while residual connections together with Layer Normalization are introduced to ensure semantic scale consistency across stages, for $i \in \{\mathrm{ART}, \mathrm{PV}, \mathrm{DL}\}$:

$$\boldsymbol{O}_i = \Phi_i\left(\sum_{s=1}^{S} \beta_i^{(s)} \left(W_i^{(s)} LSE^{(s)}(\boldsymbol{G}_i) + b_i^{(s)}\right) + \gamma_i \boldsymbol{G}_i + \delta_i \mathrm{LN}(\boldsymbol{G}_i)\right), \quad (8)$$

where $\delta_i$ is an additional residual gating scalar. To model the dynamic dependencies across different phases, the three-phase features are stacked in chronological imaging order ($\mathrm{ART} \to \mathrm{PV} \to \mathrm{DL}$), thereby constructing an additional temporal/phase dimension and forming a time-ordered sequential input. This sequence is then fed into the Spatio-Sequential Decoder (SS2D): $\boldsymbol{S} = Stack_t\left(\boldsymbol{O}_{ART}, \boldsymbol{O}_{PV}, \boldsymbol{O}_{DL}\right) \in \mathbb{R}^{T \times H' \times W' \times C'}$, $T = 3$. For each spatial position $p \in$

$\{1, \ldots, H'W'\}$, let $\boldsymbol{X}_p \in \mathbb{R}^{T \times C'}$ denote the temporal feature matrix at that location. The per-pixel multi-head temporal self-attention (SS2D) is defined as:

$$\boldsymbol{A}_p^h = \mathrm{Softmax}\left( \frac{\boldsymbol{Q}_p^h (\boldsymbol{K}_p^h)^\top + \boldsymbol{B}_p^h}{\sqrt{d_k}} + \lambda \cdot \tanh\left( \boldsymbol{Q}_p^h \boldsymbol{R} (\boldsymbol{K}_p^h)^\top \right) \right) \in \mathbb{R}^{T \times T}, \qquad (9)$$

where $\boldsymbol{Q}_p^h = \boldsymbol{X}_p \boldsymbol{W}^{Q,h}$, $\boldsymbol{K}_p^h = \boldsymbol{X}_p \boldsymbol{W}^{K,h}$, $\boldsymbol{V}_p^h = \boldsymbol{X}_p \boldsymbol{W}^{V,h}$, $\boldsymbol{B}_p^h$ is a small learned bias matrix per pixel/head, $\boldsymbol{R}$ is a learned compatibility transform and $\lambda$ a scalar coefficient.

### 4.3 QIEF-MODULE

As shown in Figure 2 (c), the QIEF-Module contains two components:

**Neighborhood Cross-Temporal Attention with Relative Position Encoding.** After processing with the LLFE-Block, a Neighbor Attention mechanism is further employed to perform cross-phase fusion between each phase feature and the global features. To enhance spatial alignment, we introduce the Rotary Position Embedding (RoPE) (S, C), which is defined as follows:

$$ROPE(x)_{2i:2i+1} = \begin{bmatrix} \cos\theta_i & -\sin\theta_i \\ \sin\theta_i & \cos\theta_i \end{bmatrix} x_{2i:2i+1}, \qquad \theta_i = 10000^{-\frac{2i}{D}}. \qquad (10)$$

For query at position $p$ and key at its neighbor $q \in \mathcal{N}(p)$, the RoPE-transformed multi-head representations are $\widetilde{\boldsymbol{Q}}_p^h = RoPE(\boldsymbol{Q}_p^h)$ and $\widetilde{\boldsymbol{K}}_q^h = RoPE(\boldsymbol{K}_q^h)$. With a learnable relative positional bias table $b_{\Delta(p,q)}^h$ (indexed by relative coordinates), the neighborhood attention weight is enhanced with a spatial-distance regularizer:

$$\boldsymbol{A}_{p,q} = \mathrm{Softmax}_{q \in \mathcal{N}(p)}\left( \frac{\widetilde{\boldsymbol{Q}}_p^h (\widetilde{\boldsymbol{K}}_q^h)^\top + b_{\Delta(p,q)}^h}{\sqrt{d_k}} + \rho \cdot \frac{\|\boldsymbol{p} - \boldsymbol{q}\|_2^2}{Z} \right), \qquad (11)$$

where $\rho$ controls the spatial penalty and $Z$ is a normalization constant. Finally, the cross-phase fusion output is obtained by weighted aggregation of neighbor values using $\boldsymbol{A}_{p,q}$.

**Residual and Channel-wise Enhancement.** The fused features $\boldsymbol{F}_{atten}$ are added to the original features via a residual connection:

$$\boldsymbol{F}_{res}^{(p)} = \alpha \cdot \left( \boldsymbol{W}_r * \boldsymbol{F}_{atten}^{(p)} + \boldsymbol{b}_r \right) + (1-\alpha) \cdot \mathrm{LN}(\widetilde{\boldsymbol{F}}^{(p)}), \quad p \in \{\mathrm{ART}, \mathrm{PV}, \mathrm{DL}\}, \qquad (12)$$

where $\alpha \in [0,1]$ is a learnable balance parameter. Subsequently, channel-wise enhancement (e.g., SE or SK modules) is introduced to perform adaptive weighting along the channel dimension: $\boldsymbol{F}_{out}^{(p)} = \left( \sigma\left(\boldsymbol{W}_2 \, \delta(\boldsymbol{W}_1 z^{(p)} + \boldsymbol{b}_1) + \boldsymbol{b}_2\right) \odot \mathrm{LayerNorm}(\boldsymbol{F}_{res}^{(p)}) \right) \otimes \gamma^{(p)}$, with $z^{(p)} = GAP(\boldsymbol{F}_{res}^{(p)}), \gamma^{(p)} = \mathrm{LayerNorm}(\boldsymbol{F}_{res}^{(p)})$. Here, GAP denotes global average pooling, $\delta$ represents the ReLU activation, and $\odot$ indicates element-wise multiplication. This operation strengthens important channel features while suppressing irrelevant background interference, thereby enhancing the discriminative capacity of the multi-phase fused representations.

## 5 EXPERIMENTS

**Dataset Curation.** We evaluate our method on two multiphase CT datasets: PLC-CECT and MPLL. ■ **PLC-CECT** (Luo et al., 2025) contains 278 patients with primary liver cancer and 83 non-cancer controls, totaling $\mathbf{152,965}$ slices ($\mathbf{50,560}$ with lesions). Four phases are available (NC, ART, PV, DL), and annotations were verified by multiple radiologists. ■ **MPLL** is an in-house dataset consisting of $\mathbf{952,601}$ 2D slices collected from 141 patients, comprising 48–777 slices per case with ART, PV, and DL phases. Images were registered using B-spline algorithms with PV as reference, and annotations were verified by three radiologists. A more detailed description is provided in Appendix B.

**Environment & Configuration.** ❶ **Environment.** All experiments were conducted on an Ubuntu 20.04 LTS server equipped with dual NVIDIA RTX 4090 GPUs (24 GB) and PyTorch 2.1.2 framework. ❷ **Training Strategy.** The training ran for 100 epochs with a batch size of 16, using SGD optimizer (initial learning rate 0.01). Data augmentation included random horizontal/vertical flips (p = 0.5). For preprocessing, ART and DL were rigidly aligned to PV slices.

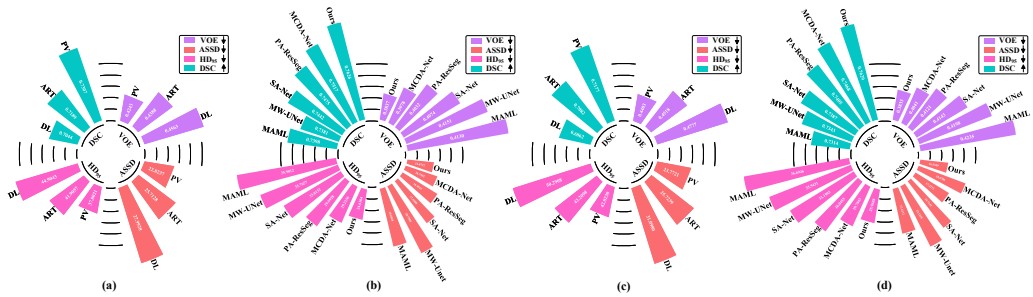

Figure 3: Segmentation performance is evaluated by four metrics: DSC, VOE, $HD_{95}$, and ASSD Subfigures show: (a) phase-specific results with clinical trend PV > ART > DL on PLC-CECT; (b) multi-phase results of `CSF-Net` on PLC-CECT; (c) phase-specific results with the same trend on MPLL; and (d) multi-phase results of `CSF-Net` on MPLL.

**Benchmarks & Evaluation.** ❶ **Datasets Configuration.** Following prior work (Jiang et al., 2023), we sample training/validation/testing splits with a ratio of 7:1:2 on the PLC-CECT and MPLL multi-phase CT benchmarks to ensure statistical consistency. ❷ **Quantitative metrics.** Dice Similarity Coefficient (DSC), Jaccard Index (JSC) for volumetric overlap, 95% Hausdorff Distance ($HD_{95}$), Average Symmetric Surface Distance (ASSD) for boundary accuracy. ❸ **Evaluation Settings. (I) Phase-wise evaluation.** where models are trained and tested on ART, PV, and DL phases separately, to reveal the inherent diagnostic value of each phase. **(II) Multi-phase evaluation.** where ART, PV, and DL phases are jointly leveraged, enabling assessment of cross-phase fusion strategies.

**Baselines & Model.** We compare against ❶ **Single-Phase Model.** ■ **Convolutional Baselines.** H-DenseUNet (Li et al., 2018), Unet++ (Zhou et al., 2019), ASSNet (Zheng et al., 2024), KiU-Net (Valanarasu et al., 2021), ■ **Transformer-based Baselines.** TransUNet (Chen et al., 2021), Swin-Unet (Cao et al., 2022). ❷ **Multi-phase Model.** MAML (Zhang et al., 2021a), MW-UNet (Zhu et al., 2022), SA-Net (Zhang et al., 2021b), PA-ResSeg (Xu et al., 2021), MCDA-Net (Kuang et al., 2024), MULLET (Wu et al., 2023). All baselines are reproduced with their official settings.

Table 1: Results on PLC-CECT dataset across ART, PV, and DL phases. Best results are **bold**, and differences relative to PV are shown as colored arrows.

| Methods | DSC(%)↑ | | | Jaccard(%)↑ | | | $HD_{95}$(mm)↓ | | | ASSD(mm)↓ | | |
|---|---|---|---|---|---|---|---|---|---|---|---|---|
| | ART | PV | DL | ART | PV | DL | ART | PV | DL | ART | PV | DL |
| H-DenseUNet (Li et al., 2018) | 71.87↓1.12 | **72.99** | 70.79↓2.20 | 56.09↓1.38 | **57.47** | 54.79↓2.68 | 41.35↑4.25 | **37.10** | 44.24↑7.14 | 24.21↑1.97 | **22.24** | 26.00↑3.76 |
| Unet++ (Zhou et al., 2019) | 69.99↓0.60 | 70.59 | 69.44↓1.15 | 53.83↓0.72 | 54.55 | 53.19↓1.36 | 46.12↑1.03 | 45.09 | 48.92↑3.83 | 27.72↑0.82 | 26.90 | 28.88↑1.98 |
| ASSNet (Zheng et al., 2024) | 68.92↓0.85 | 69.77 | 67.60↓2.17 | 52.58↓0.99 | 53.57 | 51.06↓2.51 | 49.82↑3.40 | 46.42 | 53.02↑6.60 | 29.92↑2.90 | 27.02 | 31.09↑4.07 |
| TransUNet (Chen et al., 2021) | 70.04↓1.18 | 71.22 | 69.51↓1.71 | 53.89↓1.41 | 55.30 | 53.27↓2.03 | 46.93↑4.23 | 42.70 | 47.01↑4.31 | 28.72↑3.72 | 25.00 | 29.76↑4.76 |
| KiU-Net (Valanarasu et al., 2021) | 72.14↓0.66 | 72.80 | 70.77↓2.03 | 56.42↓0.81 | 57.23 | 54.76↓2.47 | 39.98↑2.74 | 37.24 | 42.83↑5.59 | 23.13↑0.30 | 22.83 | 26.01↑3.18 |
| Swin-Unet (Cao et al., 2022) | 71.89↓1.18 | **73.07** | 70.44↓2.63 | 56.12↓1.45 | **57.57** | 54.37↓3.20 | 41.91↑4.11 | **37.80** | 44.90↑7.10 | 25.71↑2.89 | **22.82** | 27.99↑5.17 |

## 5.1 MAIN RESULTS

**Obs. ❶: Phase-wise superiority of PV.** As listed in Table 1 and Table 4, across baselines, the PV phase consistently outperforms ART and DL, e.g., Swin-Unet on PLC-CECT achieves DSC 73.07% (PV) vs. 71.89% (ART) and 70.44% (DL). This matches clinical consensus: PV provides optimal lesion, parenchyma contrast, while ART highlights hypervascular tumors with blurred boundaries, and DL suffers from contrast washout. These results reaffirm that PV> ART> DL, highlighting the need for clinically guided multi-phase integration. The corresponding single-phase experiments on the MPLL dataset are presented in Appendix C.

Table 2: Comparison of different methods on PLC-CECT and MPLL datasets. Best results are **bold**.

| Methods | PLC-CECT Dataset | | | | MPLL Dataset | | | |
|---|---|---|---|---|---|---|---|---|
| | DSC(%)↑ | Jaccard(%)↑ | $HD_{95}$(mm)↓ | ASSD(mm)↓ | DSC(%)↑ | Jaccard(%)↑ | $HD_{95}$(mm)↓ | ASSD(mm)↓ |
| MAML(Zhang et al., 2021a) | 73.98↓2.28 | 58.70↓3.74 | 35.9012↑11.27 | 18.6068↑3.93 | 73.14↓3.15 | 57.65↓4.02 | 36.6946↑10.73 | 17.6172↑1.08 |
| MW-UNet(Zhu et al., 2022) | 73.81↓2.45 | 58.49↓3.95 | 35.7657↑11.14 | 20.7588↑6.09 | 73.43↓2.86 | 58.02↓3.65 | 35.5431↑9.57 | 22.1259↑5.59 |
| SA-Net(Zhang et al., 2021b) | 74.42↓1.84 | 59.26↓3.18 | 33.0132↑8.38 | 17.4408↑2.77 | 73.87↓2.42 | 58.57↓3.10 | 33.1902↑7.22 | 19.7141↑3.18 |
| PA-ResSeg(Xu et al., 2021) | 74.75↓1.51 | 59.68↓2.76 | 29.9928↑5.36 | 16.9547↑2.28 | 74.05↓2.24 | 58.79↓2.88 | 30.0455↑4.08 | 17.2273↑0.69 |
| MCDA-Net(Kuang et al., 2024) | 75.17↓1.09 | 60.22↓2.22 | 29.2156↑4.59 | 16.7502↑2.08 | 74.68↓1.61 | 59.59↓2.08 | 29.7843↑3.82 | 18.6706↑2.14 |
| **Ours** | **76.26** | **62.44** | **24.6304** | **14.6722** | **76.29** | **61.67** | **25.9688** | **16.5349** |

**Obs. ❷: Multi-phase integration boosts performance.** Our method delivers further significant gains over prior approaches. As listed in Table 2, on PLC-CECT, MW-UNet achieves 73.81% DSC, while the recent MCDA-Net improves it to 75.17%. Our method further pushes performance to 76.26% DSC (+1.09%), 62.44% Jaccard (+2.22%), 24.63 HD$_{95}$ (↓ 4.59), and 14.67 ASSD (↓ 2.08), setting a new state-of-the-art. On MPLL, MCDA-Net obtains 74.68% DSC and 59.59% Jaccard, while our approach achieves 76.29% DSC (+1.61%) and 61.67% Jaccard (+2.08%), along with clear reductions in HD$_{95}$ and ASSD. Figure 3 provides qualitative visualizations corresponding to the quantitative results in Table 1, Table 4, and Table 2. Our method produces cleaner and more reliable segmentations across lesion scales, accurately delineating both small and large tumors. Compared to competing approaches, it avoids omissions and boundary artifacts, visually corroborating the numerical superiority reported in the tables.

These consistent improvements over the latest methods confirm that our clinically-guided spatiotemporal fusion effectively models cross-phase dynamics and yields robust segmentation gains across datasets, while significantly enhancing generalization, interpretability, and phase-robust performance in practice.

**Obs. ❸: Practical efficiency and robustness.** The superiority of the PV phase over ART and DL in single-phase experiments (Table 1) suggests that PV provides the most stable lesion–parenchyma contrast. We therefore further investigate how the propagation order influences the performance of MCQS-Branch. As shown in Table 3, all six variants outperform the baseline, confirming the benefit of sequential multi-phase modeling. However, the order design plays a crucial role: PV-dominant flows consistently yield stronger results, with **PV → ART → DL** achieving the best performance (**76.29%** DSC, **61.67%** Jaccard, lowest HD$_{95}$ and ASSD). In contrast, starting from ART or DL noticeably degrades accuracy. These findings demonstrate that propagation order is not interchangeable: anchoring sequence with PV offers both numerical gains and clinically interpretable stability.

Table 3: Ablation study on different propagation orders of the MCQS-Branch in `CSF-Net`. Best results are **bold**.

| Version | Delivery Order | | | CSF-Net Performance | | | |
|---|---|---|---|---|---|---|---|
| | Stage-1 | Stage-2 | Stage-3 | DSC(%)↑ | Jaccard(%)↑ | HD$_{95}$(mm)↓ | ASSD(mm)↓ |
| a | ART → | PV → | DL | 74.73$_{↓1.56}$ | 59.66$_{↓2.01}$ | 42.99$_{↑17.02}$ | 28.04$_{↑11.51}$ |
| b | ART → | DL → | PV | 73.88$_{↓2.41}$ | 58.58$_{↓3.09}$ | 30.84$_{↑4.87}$ | 17.79$_{↑1.25}$ |
| c | PV → | ART → | DL | **76.29** | **61.67** | **25.97** | **16.53** |
| d | PV → | DL → | ART | 76.04$_{↓0.25}$ | 61.34$_{↓0.33}$ | 37.54$_{↑11.57}$ | 23.23$_{↑6.70}$ |
| e | DL → | ART → | PV | 73.08$_{↓3.21}$ | 57.58$_{↓4.09}$ | 30.60$_{↑4.63}$ | 17.23$_{↑0.70}$ |
| f | DL → | PV → | ART | 73.68$_{↓2.61}$ | 58.33$_{↓3.34}$ | 31.39$_{↑5.42}$ | 18.26$_{↑1.73}$ |

## 5.2 QUALITATIVE ANALYSIS

Figure 4 presents qualitative comparisons on PLC-CECT (e.g. 1 and 2) and MPLL (e.g. 3 and 4) under multi-phase settings. Ground truth is shown in green, predictions in green, and discrepancies in red. Competing methods often miss small lesions or produce fragmented boundaries, especially in low-contrast regions. For instance, compared with MCDA-Net (Kuang et al., 2024), our method produces more accurate boundary delineation in the first PLC-CECT case (e.g. 1), avoiding the under-segmentation observed in competing methods. In the third MPLL example (e.g. 3), `CSF-Net` successfully captures tiny nodules that are completely missed or only partially segmented by MCDA-Net and other baselines. Across all cases, our approach achieves more complete coverage and sharper delineation, particularly for irregular tumor margins. These qualitative results corroborate the quantitative gains in Tables 1, 4, and 2, highlighting the effectiveness of clinically-guided multi-phase fusion in enhancing lesion boundary accuracy.

## 5.3 EFFICIENCY ANALYSIS

As listed in Table 5, our `CSF-Net` requires only 23.36 GFLOPs, which is over 2.3× lower than MW-UNet (53.42 GFLOPs) and up to 6.5× lower than SA-Net (152.97 GFLOPs), while achieving the **highest accuracy (76.29%)** among all methods. With a moderate parameter scale (56.20M) compared to the extremely lightweight MW-UNet (2.77$M$) and the heavy SA-Net (170.85$M$). `CSF-Net` demonstrates both **superior efficiency and performance**. This highlights the superior

computational efficiency of `CSF-Net` without compromising model capacity. Further details on the experimental setup and efficiency comparisons are provided in Appendix D.

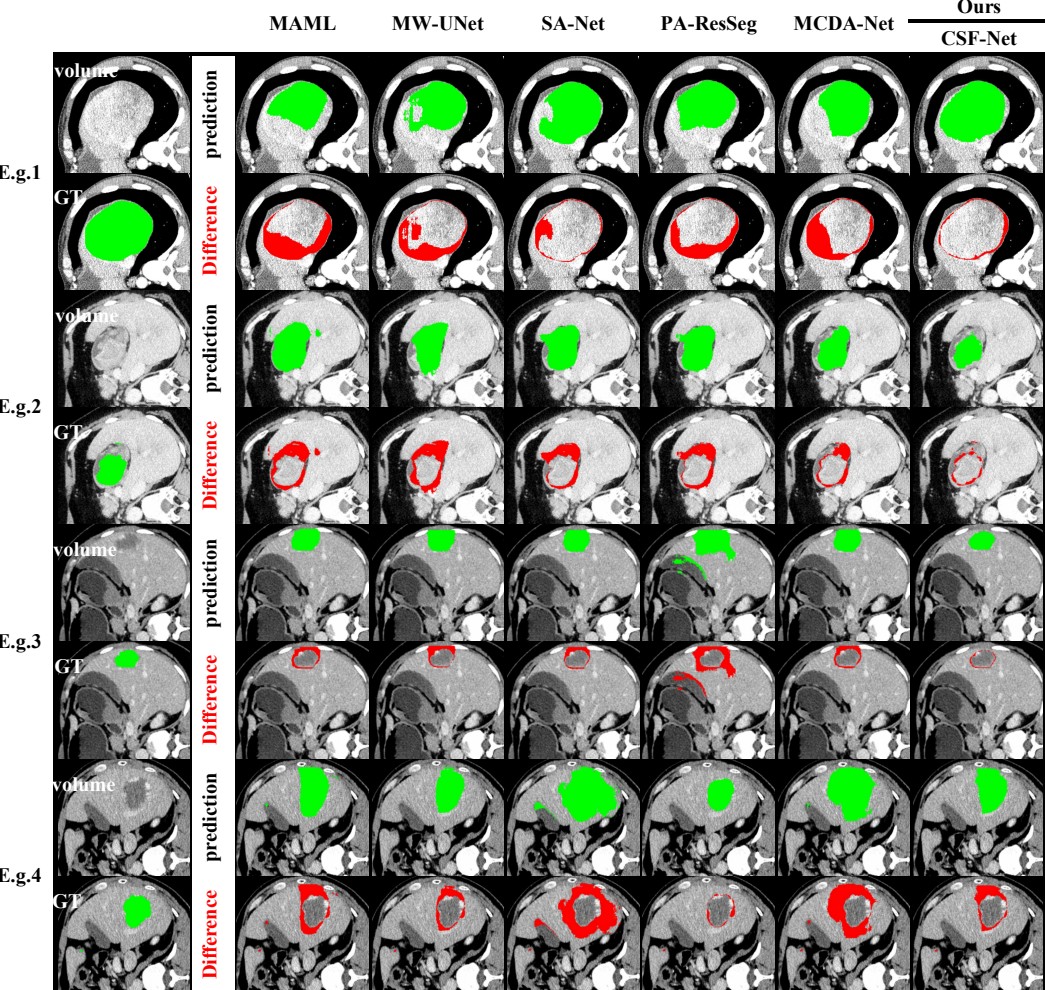

Figure 4: Qualitative comparison of `CSF-Net` with other methods. e.g.1 and 2 are from PLC-CECT and e.g.3 and 4 from MPLL. For clarity, images are cropped around lesions; green in GT marks annotated tumors, green in the prediction rows indicates segmented results, and red in the difference row highlights prediction, mask discrepancies.

## 6 CONCLUSION & LIMITATION

In this work, we conduct an in-depth investigation of multi-phase modeling and introduce `CSF-Net`, which explicitly captures cross-phase temporal dependencies by incorporating phase-aware representations and aligning them with clinically meaningful enhancement stages. Extensive experiments demonstrate that `CSF-Net` consistently outperforms strong baselines in terms of predictive accuracy and cross-phase stability, e.g., on two multi-phase CT benchmarks, `CSF-Net` surpassing the best single-phase and multi-phase baselines by $+3.22\% \uparrow$ and $+1.09\% \uparrow$, respectively. **Limitations:** While our study primarily focuses on phase-level hemodynamics, it is worth noting that digital pathology and radiomics-driven omics modalities also play vital roles in clinical oncology. In practice, integrating cross-scale information from imaging, pathology, and multi-omics often yields breakthrough insights for precision diagnosis and treatment planning. As a natural extension, we envision broadening our clinically guided spatiotemporal paradigm toward multi-modal and multi-omics fusion, further bridging radiological reasoning with pathological evidence to enable comprehensive patient-level analysis.

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

# A  ALGORITHM.

The overall training and inference procedure of `CSF-Net` is summarized in algorithm 1, where phase-specific feature encoding and cross-phase fusion are jointly integrated into a unified pipeline.

---

**Algorithm 1:** Pseudocode of `CSF-Net` (CSF-Net): multi-phase clinical-prior propagation, temporal-aware local refinement and query-interaction fusion.

---

**Input:** Multi-phase CT images $\{\boldsymbol{X}^{(ART)}, \boldsymbol{X}^{(PV)}, \boldsymbol{X}^{(DL)}\}$; Clinical graph $\mathcal{G} = (\mathcal{V}, \mathcal{E})$
      (PV,ART,DL,$\Sigma$); Ground-truth mask $Y$ (training only)

**Output:** Predicted segmentation $\hat{Y}$

**for** iteration $t \leftarrow 1$ **to** $K'$ **do**
    /* Phase-level DAG propagation (MCQS-Branch) */
    Construct $\hat{\mathcal{G}} = (\mathcal{V}, \mathcal{E})$ with $\mathcal{V} = \{\text{ART}, \text{PV}, \text{DL}, \Sigma\}$
    **for** $v_i$ in $(\mathcal{V})$ **do**
        Obtain parent set $\mathcal{P}(v_i)$ according to DAG priors
        Message aggregation: $\boldsymbol{M}^{(i)} \leftarrow (\{\boldsymbol{F}^{(j)} \mid v_j \in \mathcal{P}(v_i)\}, r_{ij}, \lambda)$
        Update node feature: $\widetilde{\boldsymbol{F}}^{(i)} \leftarrow \phi_i(\boldsymbol{F}^{(i)}, \boldsymbol{M}^{(i)})$
    **end**
    /* Local refinement and temporal ordering (TLFR-Module) */
    **for** $i \in \{\text{ART}, \text{PV}, \text{DL}\}$ **do**
        $\boldsymbol{O}_i \leftarrow (\widetilde{\boldsymbol{F}}^{(i)}; \alpha, \beta_i, \gamma_i, \delta_i)$
    **end**
    Construct temporal stack $\boldsymbol{S} =_t (\boldsymbol{O}_{ART}, \boldsymbol{O}_{PV}, \boldsymbol{O}_{DL})$
    Apply per-pixel temporal attention: $\boldsymbol{S}' \leftarrow (\boldsymbol{S}; \lambda, \boldsymbol{R}, \boldsymbol{B})$
    /* Neighborhood cross-temporal fusion (QIEF-Module) */
    **for** *spatial position* $p$ **do**
        **for** *neighbor* $q \in \mathcal{N}(p)$ **do**
            Compute RoPE-enhanced attention: $\boldsymbol{A}_{p,q} \leftarrow \left( \frac{\widetilde{\boldsymbol{Q}}_p \widetilde{\boldsymbol{K}}_q^\top + b_{\Delta(p,q)}}{\sqrt{d_k}} + \rho \frac{\|\boldsymbol{p} - \boldsymbol{q}\|_2^2}{Z} \right)$
        **end**
        Fuse: $\boldsymbol{F}_{atten}^{(p)} \leftarrow \sum_q \boldsymbol{A}_{p,q} \boldsymbol{V}_q$
    **end**
    /* Residual + Channel enhancement */
    **for** $p \in \{\text{ART}, \text{PV}, \text{DL}\}$ **do**
        $\boldsymbol{F}_{res}^{(p)} \leftarrow \alpha \cdot (\boldsymbol{F}_{atten}^{(p)}) + (1 - \alpha) \cdot (\widetilde{\boldsymbol{F}}^{(p)})$
        $\boldsymbol{F}_{out}^{(p)} \leftarrow (\boldsymbol{F}_{res}^{(p)})$
    **end**
    Aggregate output: $a^{(t)} \leftarrow (\{\boldsymbol{F}_{out}^{(p)}\})$
**end**
$\hat{Y} \leftarrow \boldsymbol{F}_{fusion}$
**if** *training* **then**
    $\mathcal{L} \leftarrow \hat{Y}, Y$ Backpropagate and update all parameters
**end**
**return** $\hat{Y}$

---

# B  THE MPLL DATASET

To enable a systematic investigation of how multi-phase contrast-enhanced CT contributes to the segmentation of liver lesions, we curated a new dataset named MPLL (Multi-Phase Liver Lesion CT). The collection originates from the "Anonymous Authoritative Hospitals (Information and ethics number will be made public after the paper is accepted)" and consisting of $952, 601$ 2D slices collected with diverse hepatic pathologies. Each individual underwent imaging in three enhancement phases (arterial, portal venous, and delayed), which collectively capture the temporal evolution of contrast and provide a comprehensive depiction of lesion appearance.

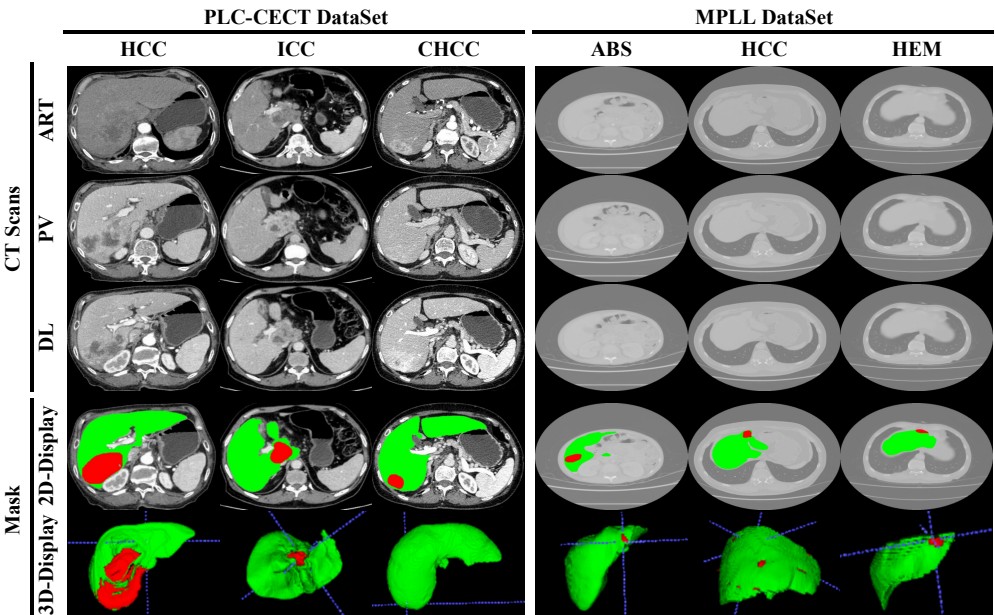

Figure 5: Example images from the PLC-CECT and MPLL datasets (green indicates liver regions, red indicates tumor regions).

❶: **Patient Cohort and Imaging Characteristics.** The retrospective cohort spans examinations performed between 2018 and 2022. Patients range in age from 9 to 72 years, reflecting both pediatric and adult populations. After anonymization to remove all personal identifiers, only imaging data were retained, and ethical approval was obtained prior to study initiation. All images share a fixed in-plane resolution of $512 \times 512$ pixels. Slice thickness varied between 0.62 mm and 5.0 mm, and scan length differed considerably among patients, yielding anywhere from 48 to 777 slices per study. Altogether, the dataset encompasses $952, 601$ axial 2D slices, making it one of the most extensive multi-phase liver lesion resources available to date.

❷: **Disease Spectrum.** The dataset includes a heterogeneous mix of liver conditions. Examples comprise hepatocellular carcinoma, intrahepatic cholangiocarcinoma, cysts, abscesses, and hemangiomas, among others. This diversity ensures that the dataset is not biased toward a single pathology, thereby offering a challenging and representative benchmark for algorithm development.

❸: **Data Partitioning.** Following practices established in recent state-of-the-art studies, MPLL was divided into training, validation, and test sets in a 7:1:2 ratio (Jiang et al., 2023). The test set was fixed to 30 cases and excluded from all model development stages to guarantee independence in evaluation and robustness in reported results.

❹: **Preprocessing and Annotation.** Because patient motion, respiration, and cardiac activity often cause misalignment across phases, we employed a B-spline registration framework, using the portal venous phase as reference. This alignment procedure reduces phase-to-phase variability and ensures spatial consistency. Lesion masks were delineated with ITK-SNAP by two experienced radiologists and subsequently reviewed by a third radiologist. This multi-stage annotation process was designed to maximize both precision and inter-observer consistency.

**Summary.** MPLL therefore offers a carefully curated, large-scale, multi-phase CT dataset with rich pathology coverage, high-quality annotations, and rigorous data partitioning. Representative examples are presented in Figure 5, illustrating the diversity of imaging phases and lesion characteristics captured.

Table 4: Results on MPLL dataset across ART, PV, and DL phases. Best results are **bold**, and differences relative to PV are shown as colored arrows.

| Methods | DSC(%)↑ | | | Jaccard(%)↑ | | | HD$_{95}$ (mm)↓ | | | ASSD (mm)↓ | | |
|---|---|---|---|---|---|---|---|---|---|---|---|---|
| | ART | PV | DL | ART | PV | DL | ART | PV | DL | ART | PV | DL |
| H-DenseUNet | $70.82_{\downarrow0.27}$ | **71.09** | $70.36_{\downarrow0.73}$ | $54.82_{\downarrow0.33}$ | **55.15** | $54.27_{\downarrow0.88}$ | $44.80_{\uparrow1.29}$ | **43.51** | $45.49_{\uparrow1.98}$ | $26.88_{\uparrow1.77}$ | **25.11** | $27.10_{\uparrow1.99}$ |
| Unet++ | $68.61_{\downarrow0.97}$ | **69.58** | $67.89_{\downarrow1.69}$ | $52.22_{\downarrow1.13}$ | **53.35** | $51.39_{\downarrow1.96}$ | $51.70_{\uparrow5.62}$ | **46.08** | $53.78_{\uparrow7.70}$ | $30.02_{\uparrow2.57}$ | **27.45** | $33.67_{\uparrow6.22}$ |
| ASSNet | $68.50_{\downarrow1.06}$ | **69.56** | $68.07_{\downarrow1.49}$ | $52.09_{\downarrow1.24}$ | **53.33** | $51.60_{\downarrow1.73}$ | $50.20_{\uparrow2.26}$ | **47.94** | $51.21_{\uparrow3.27}$ | $32.92_{\uparrow7.16}$ | **25.76** | $37.13_{\uparrow11.37}$ |
| TransUNet | $71.38_{\uparrow0.35}$ | **71.73** | $69.57_{\downarrow2.16}$ | $55.50_{\uparrow0.42}$ | **55.92** | $53.34_{\downarrow2.58}$ | $42.52_{\uparrow1.99}$ | **40.53** | $47.09_{\uparrow6.56}$ | $25.51_{\uparrow2.19}$ | **23.32** | $29.45_{\uparrow6.13}$ |
| KiU-Net | $71.24_{\downarrow0.43}$ | **71.67** | $69.83_{\downarrow1.84}$ | $55.33_{\downarrow0.48}$ | **55.85** | $53.65_{\downarrow2.20}$ | $42.31_{\uparrow0.46}$ | **41.85** | $46.25_{\uparrow4.40}$ | $25.58_{\uparrow1.07}$ | **24.51** | $28.35_{\uparrow3.84}$ |
| Swin-Unet | $70.82_{\downarrow0.95}$ | **71.77** | $68.62_{\downarrow3.15}$ | $54.82_{\downarrow1.15}$ | **55.97** | $52.23_{\downarrow3.74}$ | $43.21_{\uparrow0.39}$ | **42.82** | $50.29_{\uparrow7.47}$ | $25.72_{\uparrow1.95}$ | **23.77** | $31.10_{\uparrow7.33}$ |

## C  SINGLE-PHASE SEGMENTATION RESULTS ON THE MPLL DATASET.

As shown in table 4, across all baseline models on the MPLL dataset, the PV phase consistently yields superior segmentation performance compared to ART and DL. For instance, Swin-UNet achieves a DSC of 71.77% on PV, notably higher than 70.82% on ART and 68.62% on DL; similar trends are observed in Jaccard, HD$_{95}$, and ASSD metrics. Most models show performance drops from PV to ART and more substantial drops from PV to DL (e.g., Unet++: DSC decreases by 0.97% for ART and 1.69% for DL), while boundary-related metrics (HD$_{95}$, ASSD) worsen markedly on DL. These results align with clinical understanding: PV phase offers optimal lesion–parenchyma contrast for delineation, ART highlights hypervascular tumors but with less clear boundaries, and DL suffers from contrast washout, leading to incomplete lesion depiction. This consistent PV > ART > DL pattern underscores the necessity of leveraging clinically complementary information through multi-phase integration.

## D  DETAILED EFFICIENCY ANALYSIS.

Table 5: Efficiency Comparison of CSF-Net and Baseline Models (GFLOPs and Parameters). Performance is evaluated on the MPLL dataset under the multi-phase experiment setting. The bold indicates the best performance.

| Model | Gflops ($\times 10^9$) | Parameters (M) | Performance (%) |
|---|---|---|---|
| MAML (Zhang et al., 2021a) | 23.802 | 4.216 | 73.14 |
| MW-UNet (Zhu et al., 2022) | 53.419 | **2.773** | 73.43 |
| SA-Net (Zhang et al., 2021b) | 152.965 | 170.852 | 73.87 |
| PA-ResSeg (Xu et al., 2021) | 64.660 | 67.732 | 74.05 |
| MCDA-Net (Kuang et al., 2024) | 89.483 | 48.717 | 74.68 |
| Ours | **23.359** | 56.196 | **76.29** |

Table 5 reports the computational efficiency of `CSF-Net` compared with representative baselines. Our model achieves the lowest computational cost with only **23.359** GFLOPs, demonstrating a clear advantage over prior multi-phase segmentation networks such as MCDA-Net (89.483 GFLOPs) and SA-Net (152.965 GFLOPs). Although the parameter size of `CSF-Net` (56.196 M) is moderately larger than some lightweight baselines (e.g., MW-UNet with 2.773 M), it remains significantly smaller than heavy architectures like SA-Net (170.852 M). More importantly, `CSF-Net` delivers the best segmentation accuracy, achieving **76.29**% on the MPLL dataset, which not only surpasses lightweight designs (e.g., MAML: 73.14%, MW-UNet: 73.43%) but also consistently outperforms computationally expensive models (e.g., SA-Net: 73.87%, MCDA-Net: 74.68%). This highlights that `CSF-Net` is not only highly efficient in terms of FLOPs and parameter usage, but also achieves state-of-the-art segmentation performance, striking a favorable balance between efficiency and representational power while pushing forward the accuracy frontier.

## E  LARGE LANGUAGE MODELS USAGE STATEMENT

LLMs were used only for language polishing in this work. The manuscript was drafted entirely by the authors, and LLMs were employed solely to refine grammar and clarity of English expression.

