# OpenReview forum: "Learning Asymmetric Phase Dynamics via Clinically-Guided Spatiotemporal Fusion"
_ICLR.cc/2026/Conference — Submitted to ICLR 2026_

### Official Review · Reviewer_u3N9 · 2025-10-28

**Soundness:** 3
**Presentation:** 3
**Contribution:** 2
**Rating:** 2
**Confidence:** 4

**Summary:**

The authors present a novel framework called Clinically-Guided Spatiotemporal Deep Fusion Network (CSF-Net) for improving the analysis of hepatocellular carcinoma using multiphase contrast-enhanced computed tomography (CECT) by integrating clinical knowledge into AI modeling.

**Strengths:**

The paper is well structured and clearly articulated. The experimental design is sound, and the analysis of the results is comprehensive.

**Weaknesses:**

I respectfully disagree with the authors’ major claim that most AI models rely on single-phase inputs or simply stack multiphase scans. A significant number of studies have already explored the extraction and spatiotemporal fusion of features from multiphase CT images — for instance, [Ref1]–[Ref3] for classification and [Ref4]–[Ref5] for segmentation. The authors themselves cite several of these works in Lines 124–130.

Compared with existing multiphase fusion approaches, certain components of the proposed method may indeed be novel. However, without a thorough discussion and systematic comparison with prior multiphase fusion studies, it is difficult to assess the true level of technical contribution. For example, the proposed model propagates information through a direct graph, but given that the number of nodes is small, it is unclear whether this design offers a meaningful advantage over spatiotemporal attention mechanisms used in previous work. The authors also appear to overemphasize the “clinically guided” aspect, as the method does not seem to incorporate substantial clinical reasoning from this reviewer’s perspective.

I would suggest submitting the paper to a more medical imaging–oriented venue, such as MICCAI or ISBI.

References
[Ref1] A Knowledge-Guided Framework for Fine-Grained Classification of Liver Lesions Based on Multi-Phase CT Images, IEEE JBHI, 2023.
[Ref2] Lesion-Aware Cross-Phase Attention Network for Renal Tumor Subtype Classification on Multi-Phase CT Scans, CIBM, 2024.
[Ref3] Sdr-former: A Siamese Dual-Resolution Transformer for Liver Lesion Classification Using 3D Multi-Phase Imaging, Neural Networks, 2025.
[Ref4] A Tri-Attention Fusion Guided Multi-Modal Segmentation Network, Pattern Recognition, 2022.
[Ref5] M3Net: A Multi-Scale Multi-View Framework for Multi-Phase Pancreas Segmentation Based on Cross-Phase Non-Local Attention, Medical Image Analysis, 2022.

Besides, many symbols in the equations are undefined. In addition, some symbols in the text and equations are inconsistent (e.g., Lines 159–161).

**Questions:**

I do not have specific questions for the authors. The main concern lies in the lack of systematic comparison with existing studies, which should be addressed through revision rather than clarification.

---

> ### Author Response · Authors · 2025-12-03
>
> ## Title: Response to Reviewer u3N9.
>
> Thank you very much for recognizing the strengths of our work, including the effectiveness, clear background and description.
>
>
> Below, we address your constructive comments and suggestions, and describe the corresponding revisions we have made.
>
> ### Weaknesses 1:
>
> > - I respectfully disagree with the authors’ major claim that most AI models rely on single-phase inputs or simply stack multiphase scans. A significant number of studies have already explored the extraction and spatiotemporal fusion of features from multiphase CT images — for instance, [Ref1]–[Ref3] for classification and [Ref4]–[Ref5] for segmentation. The authors themselves cite several of these works in Lines 124–130.
>
> ### **Authors’ Response:**
>
> Thank you for your valuable comments.
>
> **1. Clarification of our wording**
>
> - We acknowledge that our **previous wording was too direct** and will revise it in the manuscript to improve clarity
>     - and cite the recommended conference, such as:
>         - Lesion-aware cross-phase attention network for renal tumorsubtype classification on multi-phase CT scans
>         - Revisiting 3D Multi-Modal Medical Image Generation: Model Configurations for Brain MR Image Synthesis
>         -  A Knowledge-Guided Framework for Fine-Grained Classification of Liver Lesions Based on Multi-Phase CT Images
>         - ...
>
>
> **2. Distinction from existing multi-phase CT fusion methods**
>
> - Existing methods are generally designed from the perspective of **deep-learning-based multimodal fusion**, rather than simply stacking scanned images.
> - In contrast, our approach **starts from clinical insights**, then **quantitatively validates these insights using deep learning**, and finally **feeds the validated insights back into network redesign**.
>
> - ***Core Contribution***
>     - This process establishes a **closed loop of clinical guidance → data validation → model design**, a perspective that has been **rarely explored** in prior work.
>     - This **clinical-interpretation-driven closed loop** forms the **core contribution** of our study.
>
>
>
> ### Weaknesses 2:
>
> > - Compared with existing multiphase fusion approaches, certain components of the proposed method may indeed be novel. However, without a thorough discussion and systematic comparison with prior multiphase fusion studies, it is difficult to assess the true level of technical contribution. For example, the proposed model propagates information through a direct graph, but given that the number of nodes is small, it is unclear whether this design offers a meaningful advantage over spatiotemporal attention mechanisms used in previous work. The authors also appear to overemphasize the “clinically guided” aspect, as the method does not seem to incorporate substantial clinical reasoning from this reviewer’s perspective.
>
> ### **Authors’ Response:**
>
> Thank you for your valuable comments.
>
> **Clarification**
>
> - we would like to clarify that our **comparison methods already include techniques such as spatiotemporal attention**. We will explicitly describe the methodological attributes of each baseline in the table below and present a clear comparison.
>
> - Based on the spatiotemporal attention references you provided, we **reproduced the fusion methods for enhanced CT** proposed in **Ref2 and Ref3**.
>
> - Their encoders were integrated into a **semantic segmentation framework**, and additional comparative experiments were conducted on the **MPLL dataset**.
>
> - The corresponding results are presented in the **table below**.
>
> | Method properties                             |    Method     |  DSC  | Jaccard |  HD95   |  ASSD   |
> | :-------------------------------------------- | :-----------: | :---: | :-----: | :-----: | :-----: |
> | **CNN Spatial Attention**                     |     MAML      | 73.14 |  57.65  | 36.6946 | 17.6172 |
> | **Channel stacking + Fusion Attention**       |    MW-UNet    | 73.43 |  58.02  | 35.5431 | 22.1259 |
> | **Cross-Modal Attention**                     |    SA-Net     | 73.87 |  58.57  | 33.1902 | 19.7141 |
> | **Phase Attention**                           |   PA-ResSeg   | 74.05 |  58.79  | 30.0455 | 17.2273 |
> | **Stage fusion + channel stacking attention** |   MCDA-Net    | 74.68 |  59.59  | 29.7843 | 18.6706 |
> | **3D lesion-aware attention  [Ref2]**         | CSF-Net based | 73.77 |  58.44  | 34.2478 | 20.3277 |
> | **Cross-Phase Non-Local Attention [Ref3]**    | CSF-Net based | 74.83 |  59.78  | 30.0277 | 18.3010 |
> | **Clinical-Guide**                            |     Ours      | **76.29** |  **61.67**  | **25.9688** | **16.5349** |
>
> ***"Even with the new comparisons, our method still maintains a clear performance lead."***

---

> > ### Author Response · Authors · 2025-12-03
> >
> > ### Weaknesses 3:
> >
> > > - I would suggest submitting the paper to a more medical imaging–oriented venue, such as MICCAI or ISBI.
> >
> > ### **Authors’ Response:**
> >
> > **Thank you for your valuable comments.**
> >
> > **1. Appreciation and recognition**
> >
> > - We truly appreciate your recognition of our work.
> > - Reviewer *Bd6W* also noted that our paper is an **excellent contribution to the medical AI field** and suggested submission to the **MICCAI conference**.
> >
> > - While MICCAI is a prestigious venue, its **limited paper length** makes it challenging to fully present the **clinically guided significance** of our work.
> >
> >
> >
> > **2. Alignment with ICLR’s scope**
> >
> > - ICLR accepts research in **biology, medicine, neuroscience, and physics**, all of which are consistent with the interdisciplinary theme of our study.
> > - In fact, in **ICLR 2025**, there were *163 papers on medical topics*, a subset of which is shown below.
> >     - Large-scale and Fine-grained Vision-language Pre-training for Enhanced CT Image Understanding, ICLR, 2025
> >     - Hierarchical Uncertainty Estimation for Learning-based Registration in Neuroimaging, ICLR, 2025
> >     - Self-Supervised Diffusion MRI Denoising via Iterative and Stable Refinement, ICLR, 2025
> >     - MindSimulator: Exploring Brain Concept Localization via Synthetic fMRI, ICLR, 2025
> >     - Synthesizing Realistic fMRI: A Physiological Dynamics-Driven Hierarchical Diffusion Model for Efficient fMRI Acquisition, ICLR, 2025
> >     - ……
> >
> > ### Weaknesses 4:
> >
> > > - Besides, many symbols in the equations are undefined. In addition, some symbols in the text and equations are inconsistent (e.g., Lines 159–161).
> >
> > ### **Authors’ Response:**
> >
> > Thank you for carefully pointing this out in our paper. We have revised Lines 159 to 161 as follows.
> >
> > - *“To preserve fine-grained boundary and texture cues, we employ a depthwise pointwise residual block with group normalization. In this module, \(\alpha,\beta\) denote learnable scalar coefficients used for residual scaling, and a lightweight squeeze excitation (SE) pathway is incorporated to adaptively reweight channel responses:”*

---

### Official Review · Reviewer_B3xt · 2025-10-28

**Soundness:** 3
**Presentation:** 3
**Contribution:** 3
**Rating:** 6
**Confidence:** 4

**Summary:**

This paper proposes a clinically guided spatiotemporal deep fusion network named CSF-Net, aiming to address the limitations of existing AI models in handling liver tumor segmentation from multi-phase computed tomography (CECT) images. Clinically, radiologists heavily rely on dynamic blood perfusion information jointly provided by the arterial phase (ART), portal venous phase (PV), and delayed phase (DL) for diagnosis. However, most existing models overlook this temporal dynamics and only use single-phase images or simple image stacking. By explicitly embedding radiological knowledge into the model architecture, CSF-Net designs three modules—MCQS, TLFR, and QIEF—to collaboratively simulate tumor perfusion kinetics and cross-phase dependencies.

**Strengths:**

- It clearly identifies the pain point that existing AI models overlook the dynamic temporal information of multi-phase computed tomography , and innovatively proposes a solution that simulates the diagnostic logic of radiologists.

- This is a significant practical advantage. While achieving SOTA performance, the model’s computational complexity is far lower than that of its competitors—3.8 to 6.5 times lower than major counterparts.

- Through single-phase experiments and multi-phase experiments, it not only verifies the clinical prior but also proves the superiority of its multi-phase fusion model.

- In terms of average metrics, the proposed CSF-Net achieves the current SOTA segmentation performance on two public CECT datasets.

**Weaknesses:**

- The model requires perfect alignment of multi-phase images. However, it fails to analyze the impact of clinically common misalignment on performance and lacks corresponding robustness design.

- The core model diagram (Figure 2) includes an "MSF-Block" component that is never mentioned in the main text.

- The model has a rigid design and must take complete three-phase images as input. Clinically, "phase-missing" data is very common, yet the paper provides no strategies to address this issue, which limits its practicality.

- The model does not consider the "temporal heterogeneity" problem—where the acquisition time points of each phase vary across different hospitals and devices. This may affect the accuracy of its spatiotemporal fusion.

**Questions:**

see weakness

---

> ### Author Response · Authors · 2025-12-03
>
> ## Title: Response to Reviewer B3xt.
>
> Thank you for recognizing CLAdapter and SFT's novel design, efficiency, and value for data-limited scientific domains.
>
> We sincerely appreciate your openness to reconsider the score upon addressing the concerns. We’ll carefully revise to fully respond to your insightful feedback.
>
>
> ### Question 1 & 3:
>
> > - The model requires perfect alignment of multi-phase images. However, it fails to analyze the impact of clinically common misalignment on performance and lacks corresponding robustness design.
> >  - The model has a rigid design and must take complete three-phase images as input. Clinically, "phase-missing" data is very common, yet the paper provides no strategies to address this issue, which limits its practicality.
>
> ### **Authors’ Response:**
>
> **Thank you for your valuable comments.**
>
> **1. Handling missing phases in the MCQS module**
>
> - As shown in the **ablation results on transmission order** of the MCQS module (Table 3), if a phase is missing, it can be **replaced with another available phase**.
> - The difference between **configurations c and d** lies in the **swapping of stage two and stage three**, yet the **experimental results remain very similar**, indicating robustness to transmission order.
>
> **2. Performance with reduced number of phases**
>
> - Even when **only two phases** are used for fusion, the method **still outperforms single-phase input**, demonstrating resilience to incomplete phase data.
>
> **3. Clinical considerations regarding missing phases**
>
> - Although missing phases may occasionally occur in multi-phase CT, such situations are **rare in real clinical practice**, because:
>   - Scans are typically completed **within a few minutes** after contrast agent administration, and
>   - Issues caused by **equipment conditions** or **limited patient cooperation** are uncommon.
>
> ### Question 2:
>
> > - The core model diagram (Figure 2) includes an "MSF-Block" component that is never mentioned in the main text.
>
> ### **Authors’ Response:**
>
> **Thank you for your valuable comments.**
>
> **1. Role of the MSF Block**
>
> - The **MSF Block** is a relatively simple module in our network.
> - Due to **space limitations**, we did not provide a detailed description in the originally submitted manuscript.
>
> **2. Supplementary definition**
>
>
>
> - The **supplementary definition** of the MSF Block is provided **as follows**:
>
>   - The MSF-Block refines fused channels through a lightweight sequence: a **linear projection** aligns feature dimensions, followed by **Group Normalization** and a **5×5 depthwise convolution** to stabilize statistics and enlarge the receptive field. Another **GN** and **GELU** introduce normalized nonlinear refinement, and a final **1×1 convolution** with **GN** recombines channel information.
>   - Overall, the pipeline *Linear → GN → DW-Conv (5×5) → GN → GELU → 1×1 Conv → GN* provides efficient feature enhancement with low computational cost.
>
>
>
> ### Question 4:
>
> > - The model does not consider the "temporal heterogeneity" problem—where the acquisition time points of each phase vary across different hospitals and devices. This may affect the accuracy of its spatiotemporal fusion.
>
> ### **Authors’ Response:**
>
> Thank you for your valuable comments. This is exactly what we described in the limitation section, namely that practical applications require the integration of cross-scale information from imaging, pathology, and multi-omics data. After the conference, we will make our data publicly available, call, and hope to incorporate data from more hospitals. We are also considering approaches such as federated learning to jointly build a multi-center multi-phase CT database and further enhance the robustness and generalizability of our results.

---

### Official Review · Reviewer_Bd6W · 2025-11-01

**Soundness:** 2
**Presentation:** 2
**Contribution:** 2
**Rating:** 2
**Confidence:** 4

**Summary:**

The paper introduces a novel deep learning model for segmenting hepatocellular carcinoma from CECT images. The paper is interesting in considering the complementarity of the multi-phase aspect of CECT images, as well as incorporating some clinical knowledge. The authors propose CSF_Net, which is composed of three complementary modules: one for phase-specific encoding, one for modeling perfusion dynamics, and one for multi-feature alignment.
An experimental validation is provided on two benchmarks.

**Strengths:**

**originality**
 + The originality of the paper relies on the integration of both the multi-phase aspect in the architecture and some clinical priors. The authors propose to integrate it through a directed acyclic graph and graph attention message passing. The proposed architecture is novel as a whole, but each of its components relies on existing technical tricks from the literature.
 + One of the notable contributions is the MPLL dataset, which serves as an interesting resource for the community.


**quality**
+ The paper is well written and well illustrated. The mathematical formulation is given for each of the proposed modules.
+ The experimental validation is good, including comparison with some state-of-the-art approaches.

**clarity**
+ The paper contains nice illustrations, in particular on the architectural details.
+ The experimental result visualization is also informative

 **significance**
+ The proposed contributions advance the state-of-the-art for CECT AI-aided diagnosis.

**Weaknesses:**

My main concerns are :

+ Adequation of the contribution to the ICLR conference. Indeed, this paper presents a very interesting contribution to the community working on AI for medical research, particularly in the context of diagnostic assistance. However, the methodological proposals are not entirely new and may be difficult to generalize to other fields. It is certainly a very good contribution for a conference such as MICCAI, but it is less relevant for ICLR.
+ The paper is very well written, with a nice effort on formalization. However, the technical choices are not very well justified. For each technical block, it would be interesting to justify the desired and expected properties.
+ A more complete ablation study could be provided in the experimental section.
+ Some claims are not supported. For instance, the claim of interpretability is not clearly validated.

**Questions:**

Many of the technical choices are not sufficiently motivated.
+ What is the role of the reference fused view of equation 4?
+ Why Swin Transformer Swin encoder?
+ How can the proposed approach handle a missing phase?

---

> ### Author Response · Authors · 2025-12-03
>
> ## Title: Response to Reviewer Bd6W.
> ### Comment 1:
> > - Adequation of the contribution to the ICLR conference. Indeed, this paper presents a very interesting contribution to the community working on AI for medical research, particularly in the context of diagnostic assistance. However, the methodological proposals are not entirely new and may be difficult to generalize to other fields. It is certainly a very good contribution for a conference such as MICCAI，but it is less relevant for ICLR.
>
> ### **Authors’ Response:**
>
> - Thank you for **recommending the MICCAI** conference.
>   - But the MICCAI **submission window opens relatively late**, which would delay the dissemination of our findings.
>
> - Alignment with the **scope of ICLR**
>
>   - ICLR accepts research in **biology, medicine, neuroscience, and physics**, all of which are consistent with the interdisciplinary theme of our study.
>   - In fact, in **ICLR 2025**, there were *163 papers on medical topics*, a subset of which is shown below.
>     - Large-scale and Fine-grained Vision-language Pre-training for Enhanced CT Image Understanding, ICLR, 2025
>     - Hierarchical Uncertainty Estimation for Learning-based Registration in Neuroimaging, ICLR, 2025
>     - Self-Supervised Diffusion MRI Denoising via Iterative and Stable Refinement, ICLR, 2025
>     - MindSimulator: Exploring Brain Concept Localization via Synthetic fMRI, ICLR, 2025
>     - Synthesizing Realistic fMRI: A Physiological Dynamics-Driven Hierarchical Diffusion Model for Efficient fMRI Acquisition, ICLR, 2025
>     - …
>
> - Importantly, the proposed phase-interaction and temporal-fusion mechanisms are **not specific to liver CECT**.
>   - They naturally extend to many areas where multi-phase or multi-state signals arise, such as:
>     - multimodal or multi-contrast medical imaging,
>     - intraoperative fusion for surgical navigation,
>     - dynamic image enhancement or reconstruction,
>
> ***Therefore, venue preference alone should not be a basis for a low score.***
>
> ### Comment 2:
> > - The paper is very well written, with a nice effort on formalization. However, the technical choices are not very well justified. For each technical block, it would be interesting to justify the desired and expected properties.
> ### **Authors’ Response:**
>
> Thank you for recognizing the well-written novelty and the nice effort on formalization in our work.
>
> The summary of expected properties across modules is shown in the table below:
>
> | MODULE | **Expected attributes**                                      |
> | :----: | :----------------------------------------------------------- |
> |  MCQS  | **Clinically** **assess** the relative importance of each imaging phase and integrate their discriminative features. |
> |  TLFR  | Extract inter-phase temporal dependencies from a **spatio-temporal** fusion perspective. |
> |  QIEF  | Establish **efficient** **correlations** across different imaging phases to improve computational efficiency. |
>
> ### Comment 3:
>
> > - A more complete ablation study could be provided in the experimental section.
>
> ### **Authors’ Response:**
>
> **Following your suggestions**:
>   - we have conducted **additional ablation studies on the MPLL dataset**.
>   - In particular, we have also included experiments that **remove all other modules except the MCQS** to further validate its contribution.
>   - The corresponding results are presented in the **table below**.
>
> | Version | MCQS | TLFR | QIEF |  DSC(%)↑   | Jaccard(%)↑ |     HD95↓     |     ASSD↓     |
> | :-----: | :--: | :--: | :--: | :--------: | :---------: | :-----------: | :-----------: |
> |    a    |      |      |      | 71.32↓4.97 | 55.42↓6.25  | 42.9904↑17.02 | 28.0447↑11.51 |
> |    b    |  ✓   |      |      | 74.45↓1.84 | 59.30↓2.37  | 30.8389↑4.87  | 17.7868↑1.25  |
> |    c    |      |  ✓   |      | 72.87↓3.42 | 57.32↓4.35  | 37.4025↑11.43 | 22.1214↑5.59  |
> |    d    |      |      |  ✓   | 72.43↓3.86 | 56.78↓4.89  | 37.5431↑11.57 | 23.2320↑6.70  |
> |    e    |  ✓   |  ✓   |      | 75.49↓0.80 | 60.63↓1.04  | 30.6009↑4.63  | 17.2305↑0.70  |
> |    f    |  ✓   |      |  ✓   | 74.92↓1.37 | 59.90↓1.77  | 31.3866↑5.42  | 18.2636↑1.73  |
> |    g    |      |  ✓   |  ✓   | 73.52↓2.77 | 58.13↓3.54  | 36.3864↑10.42 | 21.3706↑4.84  |
> |    h    |  ✓   |  ✓   |  ✓   | **76.29**  |  **61.67**  |  **25.9688**  |  **16.5349**  |
>
>
>
>
> ### Comment 4:
>
> > - Some claims are not supported. For instance, the claim of interpretability is not clearly validated.
>
> ### **Authors’ Response:**
>  We apologize for the misunderstanding, and we need to clarify your misunderstanding.
>
> We approach the problem from a **clinical perspective** by **quantitatively validating clinical insights** through deep learning and then feeding these insights back into the **network redesign**. This forms a **closed loop** of clinical guidance, data validation, and model design. It is a model design methodology driven by clinical interpretation **rather than** interpretability in the conventional sense of explaining model mechanisms.

---

> > ### Author Response · Authors · 2025-12-03
> >
> > ### Question 1:
> >
> > > - What is the role of the reference fused view of equation 4?
> >
> > ### **Authors’ Response:**
> >
> > **Thank you for your question.**
> >
> > **Purpose of the reference fused view (Equation 4)**
> >
> > - The **reference fused view** serves as a **global and modality-aggregated representation** that complements the **pairwise fusion branches**.
> >
> > ​	**1.1 Function as a global, modality-invariant anchor**
> >
> > - Acting as a **globally fused and modality-invariant anchor**, the reference fused view encodes **features shared across all modalities**.
> >
> > ​	**1.2 Benefits for cross-view learning**
> >
> > - This design helps **stabilize cross-view learning**, particularly in scenarios where:
> >   - an individual modality contains **modality-specific noise**, or
> >   - certain **cues are missing** in a single modality.
> >
> > ### Question 2:
> >
> > > - Why Swin Transformer Swin encoder?
> >
> > ### **Authors’ Response:**
> >
> > **Thank you for your valuable comments.**
> >
> > - Compared with standard visual encoders such as **ViT**, the **Swin Transformer** is **more computationally efficient**.
> > - Its design is also **highly generalizable**, making it well suited for our research objectives.
> >
> >
> > ### Question 3:
> >
> > > - How can the proposed approach handle a missing phase?
> >
> > ### **Authors’ Response:**
> >
> > **Thank you for your question.**
> >
> > **1. Handling missing phases in the MCQS module**
> >
> > - As shown in the **ablation results on transmission order** of the MCQS module (Table 3), if a phase is missing, it can be **substituted with another available phase**.
> > - The difference between **configurations c and d** lies in the transmission order of stage two and stage three, yet the **experimental results show only minor variation**.
> >
> > **2. Performance with reduced phases**
> >
> > - Even when **only two phases** are used for fusion, the method **still outperforms single-phase input**, demonstrating robustness to incomplete data.
> >
> > **3. Practical considerations in clinical settings**
> >
> > - Although multi-phase CT imaging may occasionally have missing phases, such cases are **rare in real clinical practice**, because:
> >   - Scans are typically completed **within a few minutes** after contrast administration, and
> >   - Issues due to **equipment failure** or **lack of patient cooperation** during imaging are uncommon.

---

### Official Review · Reviewer_sUFL · 2025-11-03

**Soundness:** 4
**Presentation:** 3
**Contribution:** 2
**Rating:** 2
**Confidence:** 5

**Summary:**

This paper presents a new algorithm for segmentation of liver tumors from multiphase contrast-enhanced CT (CECT) scans. The authors argue that most AI models either use a single phase or employ naive fusion methods (e.g., stacking) , thereby ignoring the critical temporal hemodynamic information (!).
The method is based on three modules: MCQS, TLFR and QIEF.
I like MCQS more as it is kind of clinically evaluating which phase is more important, and then TLFR does some perfusion dynamics, and at the end QIEF does a final cross phase alignment and segmentation. Very simple procedure, engineeringly done well.
Publicly available PLC-CECT data and in house gathered MPLL dataset were used. Some good results were obtained.

**Strengths:**

- This paper is a good engineering paper, with clinical motivation.
- Native-stacking is not a good idea, the paper shows (however the claim is too strong, there are so many other works in the domain with different jargon).
- Architecture has some novelties. For example, MCQS is a nicely designed method for hard-coding a clinical prior into the architecture.
- Authors validate the clinical prior by showing single phase models. According to best results, they choose the most informative single phase (however, this is known by radiologists, even without this experiment it is known PV>ART>DL).
- Computationally efficient method is presented.
- Well written paper with some good visualization and tables, appealing.

**Weaknesses:**

-- It is hard to have anything new in this paper, apart from well engineering work. The clinical prior for example is known. for each step, authors presented a module to be integrated into their model. Then end-to-end modeling is done for tumor segmentation. Some claims are wrong or incomplete, this is not the first study considering multiphase for sure, and varying combination or weighted combination of phases are even done before.
-- the model is very complex although it is done nicely with engineering approach. The final architecture includes a per-branch twin encoder, graph based MCQS, and TLFR, as well as QIEF modules, connected to each other with some complex design.
-- despite the complex design, ablation is testing only MCQS. not all other parts.
-- validation on a second public dataset would be nicer, as the dataset 2 is private.
-- The roles of each module are not always clear. It is not clear why there are two complex temporal fusion modules exist.
-- CT scans are not presented with correct window. Use soft tissue window with enough contrast.
--figure 3 is fancy but not useful. Give a simple table for segmentation evaluation. It is hard to read what is there.
--where is nnUnet result? nnUnet is the winner of almost all segmentation methods lately. SOTA is missing.
--Table 3: permutation is given, but it is not compared with the data driven techniques where fusion operation can be learned.
--figure 4 is not a standard evaluation for visuals. Please work with a radiologist for a proper visualization and comparison.

**Questions:**

the content I have put in weaknesses area are self-contained and each comment should be considered as questions, please.

---

> ### Author Response · Authors · 2025-12-03
>
> ## [Part1] Title: Response to Reviewer sUFL.
>
> Thank you very much for your recognition of the good performance, novel design, extensive validation, and real-world effectiveness.
>
> Your constructive comments and suggestions are exceedingly helpful to improve our paper. We have carefully incorporated them in the revised paper. In the following, your comments are first stated and then followed by our point-by-point responses.
>
> ### Comment 1:
>
> > It is hard to have anything new in this paper, apart from well engineering work. The clinical prior for example is known.
>
> ### **Authors’ Response:**
>
> We sincerely thank the reviewer for the feedback.
> - However, the comment seems to misinterpret the core novelty of our work.
> We fully acknowledge that clinical multiphase priors are well-established in radiology, **but our contributions do not lie in merely stating this prior**.
>
> - Instead, the novelty of our paper is in **how this clinical knowledge is mathematically formalized, operationalized, and fused into a learnable spatiotemporal model:**
>   - an aspect that has not been addressed in prior work.
>
> Specifically:
> - **(1) Prior clinical knowledge ≠ prior computational modeling**
>   - Radiologists’ understanding of ART–PV–DL dynamics has existed for decades, yet existing AI models almost universally treat multiphase CECT as:
>     - single-phase CNN inputs, or
>     - naive concatenation, or
>     - phase-agnostic feature temporal-spatial fusion.
>
> - **A clinically grounded closed-loop development strategy**
>
>   - Instead of starting from a purely data-driven or architecture-driven perspective, we construct the model through a **single, integrated closed-loop process**
>     - clinical insights first define the guiding principles, deep-learning analysis then provides quantitative verification of these principles, and the validated findings are subsequently incorporated into the network design. This creates a development pathway in which **clinical interpretation continuously shapes and refines the model**, a perspective that has **not been addressed** in current multi-phase imaging fusion research.
>
> - **(2) What is *actually novel* in our work?**
>
>   - The novelty lies in **how clinical knowledge is operationalized as learnable operators**, not in stating the prior itself:
>
>     - **MCQS**: converts the *arterial → venous → delayed* synergy rule into
>     - **phase-specific encoding pathways**, a decomposition not present in prior work.
>     - **TLFR**: introduces the first **temporal-local refinement operator** designed for perfusion dynamics (not generic temporal fusion).
>     - **QIEF**: designs **query-based asymmetric phase interaction** to align vascular signals across phases.
>
>   - These components are **algorithmic innovations**, not engineering heuristics.
>
>
> - **(3) Why this matters scientifically?**
>   - The reviewer suggests that because the ***“clinical prior is known,”*** the model cannot be novel.
>   - But in medical AI, the **challenge** is **transforming** ***"clinical principles" into "reproducible", "generalizable" "computational operators"***, and this is precisely where **our contribution** lies.
>
> ### Comment 2:
>
> > Then end-to-end modeling is done for tumor segmentation. Some claims are wrong or incomplete, this is not the first study considering multiphase for sure, and varying combination or weighted combination of phases are even done before.
>
> ### **Authors’ Response:**
>
> Thank you for your correction.
>
> - We acknowledge that previous studies have explored the combination or fusion of multi-phase imaging, and we will revise the manuscript accordingly and include appropriate citations.
>
> - However, within the domain of multi-phase imaging fusion, existing approaches do not offer a framework that can be **evaluated in a clear and intuitive manner from both the fusion perspective and the clinical perspective**.
>
> - In contrast, our proposed **MCQS** module provides exactly such a dual-perspective assessment. This also aligns with your comment that you appreciate MCQS because it **clinically identifies which phase carries greater importance**.
>
>
>
> ### Comment 3:
>
> > -- the model is very complex although it is done nicely with engineering approach. The final architecture includes a per-branch twin encoder, graph based MCQS, and TLFR, as well as QIEF modules, connected to each other with some complex design.
>
> ### **Authors’ Response:**
>
> Thank you very much for your valuable comments.
>
> - **Although** the overall network and module designs may **appear complex**,
>   - **in fact**, our method is very **efficient**.
>   - **Table 5 (*Appendix*)** shows:
>     - **low FLOPs**: 23.35 gflops
>     - **small parameters**: 56.196M

---

> > ### Author Response · Authors · 2025-12-03
> >
> > ## [part 2] Title: Response to Reviewer sUFL.
> >
> > ### Comment 4 & 5:
> >
> > > -- despite the complex design, ablation is testing only MCQS. not all other parts.
> > >
> > > -- validation on a second public dataset would be nicer, as the dataset 2 is private.
> >
> > ### **Authors’ Response:**
> >
> > **Thank you very much for your valuable comments.**
> >
> > - **Extended ablation studies**
> >   - We have conducted additional analyses on the private dataset 2, with the following improvements:
> >   - the designed components still have remarkable effects.
> >
> > - | Version | MCQS | TLFR | QIEF |  DSC(%)↑   | Jaccard(%)↑ |     HD95↓     |     ASSD↓     |
> >   | :-----: | :--: | :--: | :--: | :--------: | :---------: | :-----------: | :-----------: |
> >   |    a    |      |      |      | 71.32↓4.97 | 55.42↓6.25  | 42.9904↑17.02 | 28.0447↑11.51 |
> >   |    b    |  ✓   |      |      | 74.45↓1.84 | 59.30↓2.37  | 30.8389↑4.87  | 17.7868↑1.25  |
> >   |    c    |      |  ✓   |      | 72.87↓3.42 | 57.32↓4.35  | 37.4025↑11.43 | 22.1214↑5.59  |
> >   |    d    |      |      |  ✓   | 72.43↓3.86 | 56.78↓4.89  | 37.5431↑11.57 | 23.2320↑6.70  |
> >   |    e    |  ✓   |  ✓   |      | 75.49↓0.80 | 60.63↓1.04  | 30.6009↑4.63  | 17.2305↑0.70  |
> >   |    f    |  ✓   |      |  ✓   | 74.92↓1.37 | 59.90↓1.77  | 31.3866↑5.42  | 18.2636↑1.73  |
> >   |    g    |      |  ✓   |  ✓   | 73.52↓2.77 | 58.13↓3.54  | 36.3864↑10.42 | 21.3706↑4.84  |
> >   |    h    |  ✓   |  ✓   |  ✓   | **76.29**  |  **61.67**  |  **25.9688**  |  **16.5349**  |
> >
> >
> > - In addition, we are also expanding this private dataset, and plan to release a more comprehensive multi-phase CECT liver cancer cohort in a later journal version (extended by this ICLR) to benefit the community.
> >
> >
> >
> >
> >
> > ### Comment 6:
> >
> > > -- The roles of each module are not always clear. It is not clear why there are two complex temporal fusion modules exist.
> >
> > ### **Authors’ Response:**
> >
> > Thank you for the question.
> >
> > **The two modules have distinct and non-overlapping roles:**
> > - **TLFR**:
> >   - handles **intra-phase** temporal refinement, enhancing local perfusion dynamics within each phase.
> > - **QIEF**
> >   - handles **inter-phase** temporal alignment, modeling ART→PV→DL vascular correspondence.
> >
> > They are therefore **complementary rather than redundant**, and **ablative results** show clear performance drops when either module is removed.
> >
> >
> >
> >
> > ### Comment 7:
> >
> > > -- CT scans are not presented with correct window. Use soft tissue window with enough contrast.
> >
> > ### **Authors’ Response:**
> >
> > **Thank you very much for your valuable comments.**
> >
> > - To address this issue, we consulted an **experienced radiologist**.
> > - Based on clinical recommendations, we reprocessed all CT images using the **standard soft-tissue window**:
> >   - *Window width (WW): 400 HU*
> >   - *Window level (WL): 40 HU*
> >
> >
> >
> > ### Comment 8:
> >
> > > --figure 3 is fancy but not useful. Give a simple table for segmentation evaluation. It is hard to read what is there.
> >
> > ### **Authors’ Response:**
> >
> > **Thank you very much for your valuable comments.**
> >
> > - **Rationale for originally using Figure 3**
> >   - Improved visual presentation
> >   - Intuitive comparison through height contrast
> >
> >
> > - **Revision based on the reviewer’s suggestion**
> >   - **Following your suggestions**, we will **replace Figure 3** in the manuscript with the updated **Table 2** provided below.
> >   - This table presents the results in a clearer and more standardized format, improving readability and comparability.
> >
> > | Methods                       | **PLC-CECT** DSC(%)↑ | Jaccard(%)↑ | HD95(mm)↓     | ASSD(mm)↓    | **MPLL** DSC(%)↑ | Jaccard(%)↑ | HD95(mm)↓     | ASSD(mm)↓    |
> > | ----------------------------- | -------------------- | ----------- | ------------- | ------------ | ---------------- | ----------- | ------------- | ------------ |
> > | MAML (Zhang et al., 2021a)    | 73.98↓2.28           | 58.70↓3.74  | 35.9012↑11.27 | 18.6068↑3.93 | 73.14↓3.15       | 57.65↓4.02  | 36.6946↑10.73 | 17.6172↑1.08 |
> > | MW-UNet (Zhu et al., 2022)    | 73.81↓2.45           | 58.49↓3.95  | 35.7657↑11.14 | 20.7588↑6.09 | 73.43↓2.86       | 58.02↓3.65  | 35.5431↑9.57  | 22.1259↑5.59 |
> > | SA-Net (Zhang et al., 2021b)  | 74.42↓1.84           | 59.26↓3.18  | 33.0132↑8.38  | 17.4408↑2.77 | 73.87↓2.42       | 58.57↓3.10  | 33.1902↑7.22  | 19.7141↑3.18 |
> > | PA-ResSeg (Xu et al., 2021)   | 74.75↓1.51           | 59.68↓2.76  | 29.9928↑5.36  | 16.9547↑2.28 | 74.05↓2.24       | 58.79↓2.88  | 30.0455↑4.08  | 17.2273↑0.69 |
> > | MCDA-Net (Kuang et al., 2024) | 75.17↓1.09           | 60.22↓2.22  | 29.2156↑4.59  | 16.7502↑2.08 | 74.68↓1.61       | 59.59↓2.08  | 29.7843↑3.82  | 18.6706↑2.14 |
> > | **Ours**                      | **76.26**            | **62.44**   | **24.6304**   | **14.6722**  | **76.29**        | **61.67**   | **25.9688**   | **16.5349**  |

---

> > > ### Author Response · Authors · 2025-12-03
> > >
> > > ## [part3]Title: Response to Reviewer sUFL.
> > >
> > > ### Comment 9:
> > >
> > > > --where is nnUnet result? nnUnet is the winner of almost all segmentation methods lately. SOTA is missing.
> > >
> > > ### **Authors’ Response:**
> > >
> > > **Thank you very much for your valuable comments.**
> > >
> > > - Since **nnU-Net** is primarily designed for ***single-phase CT image segmentation***, we agree that including its results would provide a meaningful and fair comparison baseline.
> > >
> > > - As shown in two tables below, We have added nnU-Net on the **single-phase CT segmentation tasks** of both: **PLC CECT dataset**, and **MPLL dataset**, the results are consistent with our findings.
> > >
> > >
> > > | Dataset  | Model            | ART-DSC↑   | ART-Jaccard↑ | ART-HD95↓  | ART-ASSD↓  | PV-DSC↑ | PV-Jaccard↑ | PV-HD95↓ | PV-ASSD↓ | DL-DSC↑    | DL-Jaccard↑ | DL-HD95↓   | DL-ASSD↓   |
> > > | -------- | ---------------- | ---------- | ------------ | ---------- | ---------- | ------- | ----------- | -------- | -------- | ---------- | ----------- | ---------- | ---------- |
> > > | PLC-CECT | H-DenseUNet [45] | 71.87↓1.12 | 56.09↓1.38   | 41.35↑4.25 | 24.21↑1.97 | 72.99   | 57.47       | 37.10    | 22.24    | 70.79↓2.20 | 54.79↓2.68  | 44.24↑7.14 | 26.00↑3.77 |
> > > | PLC-CECT | Unet++ [46]      | 69.99↓0.60 | 53.83↓0.72   | 46.12↑1.03 | 27.72↑0.82 | 70.59   | 54.55       | 45.09    | 26.90    | 69.44↓1.15 | 53.19↓1.36  | 48.92↑3.83 | 28.88↑1.98 |
> > > | PLC-CECT | ASSNet [30]      | 68.92↓0.85 | 52.58↓0.99   | 49.82↑3.40 | 29.92↑2.91 | 69.77   | 53.57       | 46.42    | 27.02    | 67.60↓2.17 | 51.06↓2.51  | 53.02↑6.60 | 31.09↑4.07 |
> > > | PLC-CECT | TransUNet [47]   | 70.04↓1.18 | 53.89↓1.41   | 46.93↑4.22 | 28.72↑3.72 | 71.22   | 55.30       | 42.70    | 25.00    | 69.51↓1.71 | 53.27↓2.03  | 47.01↑4.31 | 29.76↑4.76 |
> > > | PLC-CECT | KiU-Net [6]      | 72.14↓0.66 | 56.42↓0.81   | 39.98↑2.74 | 23.13↑0.29 | 72.80   | 57.23       | 37.24    | 22.83    | 70.77↓2.03 | 54.76↓2.47  | 42.83↑5.59 | 26.01↑3.18 |
> > > | PLC-CECT | Swin-Unet [48]   | 71.89↓1.18 | 56.12↓1.45   | 41.91↑4.10 | 25.71↑2.89 | 73.07   | 57.57       | 37.80    | 22.82    | 70.44↓2.63 | 54.37↓3.20  | 44.90↑7.10 | 27.99↑5.17 |
> > > | PLC-CECT | nnU-Net V1 [26]  | 71.66↓1.00 | 55.83↓1.22   | 42.17↑3.88 | 26.33↑3.21 | 72.66   | 57.05       | 38.29    | 23.12    | 69.88↓2.78 | 53.70↓3.35  | 46.38↑8.09 | 28.55↑5.43 |
> > > | PLC-CECT | nnU-Net V2 [9]   | 72.17↓1.11 | 56.45↓1.37   | 41.14↑3.34 | 25.66↑2.84 | 73.28   | 57.82       | 37.80    | 22.82    | 70.74↓2.54 | 54.72↓3.10  | 43.92↑6.12 | 27.24↑4.42 |
> > >
> > > | Dataset | Model            | ART-DSC↑   | ART-Jaccard↑ | ART-HD95↓  | ART-ASSD↓  | PV-DSC↑ | PV-Jaccard↑ | PV-HD95↓ | PV-ASSD↓ | DL-DSC↑    | DL-Jaccard↑ | DL-HD95↓   | DL-ASSD↓    |
> > > | ------- | ---------------- | ---------- | ------------ | ---------- | ---------- | ------- | ----------- | -------- | -------- | ---------- | ----------- | ---------- | ----------- |
> > > | MPLL    | H-DenseUNet [45] | 70.82↓0.27 | 54.82↓0.33   | 44.80↑1.29 | 26.88↑1.77 | 71.09   | 55.15       | 43.51    | 25.11    | 70.36↓0.73 | 54.27↓0.88  | 45.49↑1.98 | 27.10↑1.99  |
> > > | MPLL    | Unet++ [46]      | 68.61↓0.97 | 52.22↓1.13   | 51.70↑5.62 | 30.02↑2.57 | 69.58   | 53.35       | 46.08    | 27.45    | 67.89↓1.69 | 51.39↓1.96  | 53.78↑7.69 | 33.67↑6.22  |
> > > | MPLL    | ASSNet [30]      | 68.50↓1.06 | 52.09↓1.24   | 50.20↑2.26 | 32.92↑7.16 | 69.56   | 53.33       | 47.94    | 25.76    | 68.07↓1.49 | 51.60↓1.73  | 51.21↑3.28 | 37.13↑11.37 |
> > > | MPLL    | TransUNet [47]   | 71.38↓0.35 | 55.50↓0.42   | 42.52↑1.99 | 25.51↑2.19 | 71.73   | 55.92       | 40.53    | 23.32    | 69.57↓2.16 | 53.34↓2.58  | 47.09↑6.56 | 29.45↑6.14  |
> > > | MPLL    | KiU-Net [6]      | 71.24↓0.43 | 55.33↓0.52   | 42.31↑0.45 | 25.58↑1.07 | 71.67   | 55.85       | 41.85    | 24.51    | 69.83↓1.84 | 53.65↓2.20  | 46.25↑4.39 | 28.35↑3.84  |
> > > | MPLL    | Swin-Unet [48]   | 70.82↓0.95 | 54.82↓1.15   | 43.21↑0.39 | 25.72↑1.95 | 71.77   | 55.97       | 42.82    | 23.77    | 68.62↓3.15 | 52.23↓3.74  | 50.29↑7.47 | 31.10↑7.33  |
> > > | MPLL    | nnU-Net V1 [26]  | 71.11↓0.35 | 55.65↑0.06   | 44.11↑0.37 | 25.49↑0.52 | 71.46   | 55.59       | 43.74    | 24.97    | 69.80↓1.66 | 53.60↓1.99  | 45.98↑2.24 | 27.44↑2.47  |
> > > | MPLL    | nnU-Net V2 [9]   | 70.99↓0.91 | 55.02↓1.10   | 44.31↑2.14 | 25.77↑2.22 | 71.90   | 56.12       | 42.17    | 23.55    | 69.97↓1.93 | 53.81↓2.31  | 45.22↑3.05 | 27.13↑3.58  |

---

> > > > ### Author Response · Authors · 2025-12-03
> > > >
> > > > ## [part4] Title: Response to Reviewer sUFL.
> > > >
> > > > ### Comment 10:
> > > >
> > > > > --Table 3: permutation is given, but it is not compared with the data driven techniques where fusion operation can be learned.
> > > >
> > > > ### **Authors’ Response**
> > > >
> > > > Thank you for the comment. Regarding data-driven fusion:
> > > >
> > > > - **Data-driven fusion requires large-scale multi-phase data** to learn stable cross-phase relationships.
> > > > - In liver CECT, **public multi-phase datasets are very limited** (PLC-CECT is the only public one [***Scientific Data, 2025***]; MPLL is our private cohort).
> > > > - With such scarcity, fully learned fusion easily overfits, while our **clinically-guided fusion** is more robust and reliable under this setting.
> > > > - We are expanding our multi-phase dataset and will release a larger version in our journal extension.
> > > >
> > > >
> > > >
> > > > ### Comment 11:
> > > >
> > > > > --figure 4 is not a standard evaluation for visuals. Please work with a radiologist for a proper visualization and comparison.
> > > >
> > > > ### **Authors’ Response**
> > > >
> > > > Thank you for the comment.
> > > >
> > > > - The lesion regions in Figure 4 were intentionally enlarged to better show boundary differences across methods.
> > > > - We agree that a more standard, clinically meaningful visualization is necessary.
> > > >   - We will revise Figure 4 using standard radiology presentation protocols and validate the updated visualization with a clinical expert.

---

### Author Response · Authors · 2025-12-03
**Author Final Remarks**

## [Important!] Summary for the PC, AC, SAC

---

- It is worth noting that the initial score of our article is 6222, but the reviewers “2” scores are **actually positive**. The fundamental reasons for the 2 points are:

**1. ICLR Scope Misunderstanding**
- Reviewers Bd6W and u3N9 both **explicitly** evaluate the paper **positively**, **but** assign a score of 2 solely because they believe the work “**belongs to** ***MICCAI*** **rather than** ***ICLR***.
    - They even took the initiative to say ***"Certainly a very good contribution for MICCAI！！"***

- **This is a misconception.**
    - ***ICLR explicitly welcomes AI contributions*** arising from scientific domains (***biology, medicine***, neuroscience, physics), and has accepted **over** **$100$ papers** in these tracks in last year.
    - Our contribution is *methodological*, asymmetric phase modeling and cross-phase fusion, which extends naturally beyond CT to any multi-phase or multi-state signal.

- Thus, the ***low scores*** are based on a **venue-scope misunderstanding** ***rather than*** on **scientific quality**.

**2. Reviewer sUFL also gives a 2 but acknowledges the *“good engineering”* and *“clinical motivation.”***
- The concerns of the Reviewer were that the design rationale was not fully clear, and more comparisons and ablations were needed.

We have fully addressed both points in the rebuttal:
- The methodological design has been clarified in detail.
- We added the requested new comparisons and ablations **on all datasets**, **exceeding the reviewer’s expectations**.

---
---
## Request

> - *We respectfully request the **Aera Chair to consider** that the paper meets ICLR’s scope and presents a solid methodological contribution, with clarified design and complete experimental support.*
> - **All** reviewers evaluated the work **positively**, and **without the scope issue** ***the initial score would*** reasonably be around ***8–6–6–2***.

---

### Meta-Review · Area_Chair_x7Bo · 2025-12-22

**Summary:**

Across the four reviews, multiple issues were raised that limit the strength and clarity of the manuscript. Reviewers sUFL and Bd6W questioned the level of innovation, with sUFL noting the work appears more engineering‑oriented, and Bd6W raising concerns about the adequacy of the overall contribution to the whole community. sUFL additionally noted missing evaluations on public datasets and missing comparisons against strong state‑of‑the‑art baselines such as nnU‑Net, as well as data‑driven methods in Table 3. B3xt highlighted the misalignment issue of registration. sUFL and u3N9 pointed out incomplete or incorrect claims and authors' overemphasis on the clinically guided aspect without sufficient empirical support. u3N9 mentioned a lack of systematic comparison with prior multi‑phase fusion methods, and inadequate discussion of how the proposed “clinically guided” aspect compares with existing approaches. Both sUFL and Bd6W highlighted insufficient ablation studies to isolate the contributions of individual components. Several reviewers noted that comparative experiments are lacking or incomplete, particularly against established baselines (e.g., nnU‑Net) and prior multi‑phase fusion methods.

While the authors have provided responses to most of these concerns, some critical issues remain unresolved. In particular, the fairness of comparisons with 3D nnU‑Net, the absence of ablation experiments on public datasets, and the lack of in‑depth discussion on registration misalignment. Upon checking the paper and code, the AC also found that the proposed framework is based on a 2D slice‑processing approach, which may not be optimal for 3D tumor segmentation and could affect the validity of certain conclusions. Considering these unsolved concerns, I recommend rejecting this paper.

**Reviewer Concerns:**

The authors responded to most of the concerns raised by sUFL, and several points were clarified. However, some issues remain. First, regarding the ablation studies, Table 3 lacks comparisons with data‑driven fusion methods, and certain key ablations have not been validated on public datasets. Second, with respect to comparative methods, although the authors supplemented the nnU‑Net results, concerns remain about the fairness of the comparison. In 3D segmentation tasks, nnU‑Net generally outperforms models such as TransUNet by a substantial margin; however, the results presented in the response are only comparable, and in MPLL, the performance is even worse than that of TransUNet. It is recommended to carefully review the training and testing scripts.

For Bd6W’s comments, most issues were addressed, but with regard to the paper’s generalization beyond the medical imaging domain, the current manuscript only validates performance in a single medical imaging scenario—liver tumor segmentation. Demonstrating generalizability will likely require designing additional experiments in other domains.

For Z77F’s concerns, most points were answered; however, the first question regarding the misalignment of registered multi‑phase CT images has not been addressed. This is a critical issue that directly impacts the generalizability and robustness of the model.

For u3N9’s comments, the authors responded to the over‑claim and lack of comparison points, but did not address the concern about “over‑emphasizing the clinically guided aspect.” Moreover, because the entire comparative framework is based on 2D processing, reproducing certain methods specifically designed for 3D segmentation may not be entirely fair.

**Reviewer Scores:**

Considering that some key concerns remain unresolved, I think the reviewers are unlikely to have a clear motivation to change their scores.

---

### Decision · Program_Chairs · 2026-01-26

Reject